# Long-read sequencing unveils high-resolution HPV integration and its oncogenic progression in cervical cancer

Liyuan Zhou[1,10], Qiongzi Qiu [2,10], Qing Zhou[2,10], Jianwei Li[2], Mengqian Yu[1], Kezhen Li [3], Lingling Xu[1], Xiaohui Ke[4], Haiming Xu [5], Bingjian Lu[2,6], Hui Wang[2,6], Weiguo Lu [6,7✉], Pengyuan Liu[1,6,8,9✉] & Yan Lu [2,6,8✉]

Integration of human papillomavirus (HPV) DNA into the human genome is considered as a key event in cervical carcinogenesis. Here, we perform comprehensive characterization of large-range virus-human integration events in 16 HPV16-positive cervical tumors using the Nanopore long-read sequencing technology. Four distinct integration types characterized by the integrated HPV DNA segments are identified with Type B being particularly notable as lacking *E6/E7* genes. We further demonstrate that multiple clonal integration events are involved in the use of shared breakpoints, the induction of inter-chromosomal translocations and the formation of extrachromosomal circular virus-human hybrid structures. Combined with the corresponding RNA-seq data, we highlight *LINC00290*, *LINC02500* and *LENG9* as potential driver genes in cervical cancer. Finally, we reveal the spatial relationship of HPV integration and its various structural variations as well as their functional consequences in cervical cancer. These findings provide insight into HPV integration and its oncogenic progression in cervical cancer.

[1] Key Laboratory of Precision Medicine in Diagnosis and Monitoring Research of Zhejiang Province, Sir Run Run Shaw Hospital, Zhejiang University School of Medicine, Hangzhou, China. [2] Zhejiang Provincial Key Laboratory of Precision Diagnosis and Therapy for Major Gynecological Diseases, Women's Hospital, Zhejiang University School of Medicine, Hangzhou, China. [3] Department of Obstetrics and Gynecology, Tongji Hospital, Tongji Medical College, Huazhong University of Science and Technology, Wuhan, China. [4] Department of Obstetrics and Gynecology, Wenzhou Central Hospital, Wenzhou, Zhejiang 325000, China. [5] Institute of Bioinformatics, Zhejiang University, Hangzhou, China. [6] Cancer center, Zhejiang University, Hangzhou, China. [7] Women's Reproductive Health Key Laboratory of Zhejiang Province, Women's Hospital, Zhejiang University School of Medicine, Hangzhou, China. [8] Institute of Translational Medicine, Zhejiang University School of Medicine, Hangzhou, China. [9] Department of Physiology and Center of Systems Molecular Medicine, Medical College of Wisconsin, Milwaukee, WI, USA. [10] These authors contributed equally: Liyuan Zhou, Qiongzi Qiu, Qing Zhou. ✉email: lbwg@zju.edu.cn; pyliu@zju.edu.cn; yanlu76@zju.edu.cn

Cervical cancer has the second-highest incidence rate and mortality rate of all gynecological malignancies worldwide, accounting for ~604,000 new cases and 342,000 deaths in 2020[1]. Persistent infection with high-risk human papillomavirus (HPV) is the main cause of cervical cancer[2,3]. HPV incorporates a group of small (~7.9 kb in size) circular double-stranded DNA viruses that commonly infect humans, often showing considerable tropism on cutaneous and mucosal epithelia. It consists mainly of three functional parts: an early region that encodes for proteins of *E1, E2, E4–E7* genes required for viral replication; a late region that encodes for proteins of *L1* and *L2* genes necessary for viral assembly; and an upstream regulatory region (URR) between *E6* and *L1*, important for viral gene transcription and DNA replication. To date, more than 200 types of HPV have been identified[4,5], of which 15 are assigned as high-risk types because of their association with carcinogenicity[6,7]. HPV16 is the most predominant high-risk type detected in cervical diseases, followed by HPV18, which together contribute to over 70% of cervical cancers worldwide[8,9].

Most HPV infections are transient and can be cleared spontaneously by the immune system within 1–2 years. However, a few persist for many years or even decades, and eventually lead to cancer[10,11]. Contrary to the episomal state during the normal infectious cycle, high-risk HPV DNA is often integrated into the host genome during persistent infections. This is considered an important molecular event in cervical carcinogenesis[12–14]. Upon integration, deregulated expression of viral oncogenes *E6* and *E7* is frequently observed, which may result from the disruption of their negative regulators E1 and/or E2[15], and the increased stability of E6 and E7 mRNAs expressed from the viral-host fusion transcripts[16,17]. Of note, a study also showed that there was no significant association between the expression levels of viral oncogenes and the physical state of the viral genome based on cervical biopsy materials, implying that the constitutive expression rather than the expression level of viral oncogenes seems to play a decisive role in the transformation and the maintenance of malignant phenotype[18]. Oncoproteins E6 and E7 are thought to be major contributors to cervical malignant transformation by interfering with a variety of key cellular molecules, including the inactivation of the cellular tumor suppressors p53 and pRb, respectively[19–21]. Concomitants to this process are accumulated genetic alterations (such as genomic instability and structural variation) in the host genome, which is another frequently observed event associated with HPV integration and represents important co-factors in tumorigenesis[22–24]. Notably, it has been argued that enhanced expression of viral oncogenes per se is sufficient for cellular immortalization but not for malignant transformation and that secondary genetic events centered on the host genome are also necessary for malignant progression of HPV-related cancers[19,23].

Next-generation sequencing (NGS) technologies can process millions to billions of short reads (~150–300 bp) in parallel over short time periods. This has revolutionized genomic and biomedical research. Using NGS technologies, several studies have been carried out to characterize genome-wide alterations in both viral and host genomes upon HPV integration in cervical cancer[25–31]. Although the integration sites were dispersed across the human genome, several recurrent genomic hotspots that are potentially involved in important cancer-related pathways were also identified[26–28]. Likewise, viral breakpoints also occurred throughout the viral genome, although these were more likely to occur in *E1* or *E2*[25–27]. The coexistence of scattered breakpoints and recurrent hotspots could be the combined result of genomic accessibility (such as CpG regions, common fragile sites, etc.)[32,33] and selective advantage[27]. HPV integration can trigger rearrangements and amplification of the host genome, especially changing (most likely increasing) the expression of host genes around integration sites[26–28]. Several integration mechanisms have been proposed, two of which are widely accepted. In the first, a microhomology-mediated DNA repair model is suggested based on the frequent observation of microhomologous sequence between the host and viral genomes near the integration site[27]. In the second, the "looping" model mediated by DNA replication and recombination is proposed to explain the HPV-host integrant concatemers at integration sites though it is not experimentally verified[25].

One drawback of the short-read length (~150–300 bp) of the current NGS technologies is that this limits further characterization of HPV integration on the genome. It only allows detection of integration sites (i.e., breakpoints) between virus and human genomes, rather than complete clonal integration events that incorporate the collection of chimeric reads (>5 kb) containing at least one fragment of HPV DNA surrounded by the human genome on both sides. In some short-read NGS studies, integration site and integration event were used interchangeably to express the same molecular event, that is, the integration site was taken for granted as an integration event[34,35]. In other studies, two or more close breakpoints (about hundreds of kb apart) were inferred as an integration event[25,27]. Due to the heterogeneity of tumor cells, these integration breakpoints were not necessarily from the same clone. Therefore, the integration events defined by these discrete breakpoints may not represent a single clonal integration event in tumors. In addition, it is difficult to identify long or complex structural variations in the host genome upon HPV integration using the current short-read NGS technologies.

To address the above challenges, in this study, we perform a comprehensive characterization of large-range virus-human integration events in 16 HPV16-positive primary cervical tumors using the Nanopore long-read sequencing technology. We identify a total of 63 HPV integration breakpoints in 15 cervical tumors, of which 12 are identified with at least one clonal integration event. Four distinct integration types that are frequently present in the identified clonal integration events are characterized. We reveal that multiple clonal integration events are involved in the use of shared breakpoints. Moreover, we demonstrate the induction of HPV-mediated translocations between chromosomes and the formation of extrachromosomal circular (ECC) human-virus hybrid DNA structures in these clonal integration events. Combined with the corresponding RNA-seq data, we highlight *LINC00290, LINC02500,* and *LENG9* as potential driver genes for cervical cancer. Finally, we reveal the spatial relationship of HPV integration and its various structural variations (SVs) as well as the functional consequences of these SVs in cervical cancer. These findings provide insight into the HPV integration and its oncogenic progression in cervical cancer.

## Results

**Characterization of integration breakpoints.** To gain insights into HPV integration, a Nanopore long-read genome sequencing was performed in 16 HPV16-positive primary cervical carcinomas. On average, we generated 102.1 Gb of sequences per sample with a median read length of 10.9 kb and a mean base quality of 9.3 (Supplementary Table 1). Using the long-read mapper NGMLR[36], the cleaned long reads were aligned to a custom in-house reference genome comprised of the human genome and multiple HPV genomes. Sniffles, an SV caller for long-read sequences[36], was then employed to detect virus-human integration breakpoints with at least three supporting chimeric reads. As a result, a total of 63 integration breakpoints were identified in 15 out of 16 cervical tumors (Table 1 and Supplementary Table 2).

**Table 1 Characteristics of the 16 tumor samples and the number of breakpoints and integration events detected in these samples.**

| Patient-ID | HPV type | Stage | Breakpoints | Events | E6 | E7 | Follow-up (months) | Status at the end of follow-up |
|---|---|---|---|---|---|---|---|---|
| ZLR-01 | 16 | stage I | 2 | 1 | 2.9 | 8.1 | 49 | tumor-free |
| ZLR-02 | 16 | stage III | 2 | 1 | 140.8 | 460.2 | 30 | tumor-free |
| ZLR-03 | 16 | stage II | 3 | 1 | 67.5 | 177.5 | 28 | tumor-free |
| ZLR-04 | 16 | stage I | 3 | 1 | 91.4 | 212.8 | 63 | tumor-free |
| ZLR-05 | 16 | stage II | 0 | 0 | 52.3 | 146.4 | 51 | tumor-free |
| ZLR-06 | 16 | Stage II | 13 | 2 | 320.5 | 683.4 | 88 | tumor-free |
| ZLR-07 | 16 | Stage I | 2 | 0 | 115.1 | 236.4 | 90 | tumor-free |
| ZLR-08 | 16 | Stage I | 12 | 5 | 504.3 | 1213.7 | NA | NA |
| ZLR-09 | 16 | Stage II | 2 | 0 | 703.9 | 1261.5 | 61 | tumor-free |
| ZLR-10 | 16 | Stage I | 1 | 0 | 12.9 | 20.3 | 20 + 44 | relapse + tumor-free |
| ZLR-11 | 16 | Stage II | 10 | 4 | 600.1 | 1251.1 | 14 + 3 | relapse + deceased |
| ZLR-12 | 16 | Stage II | 4 | 3 | 271.6 | 539.2 | 54 + 18 | relapse + deceased |
| ZLR-13 | 16 | Stage II | 3 | 1 | 777.2 | 1719.0 | 90 | tumor-free |
| ZLR-14 | 16 | Stage II | 2 | 1 | 162.9 | 285.6 | 60 | tumor-free |
| ZLR-15 | 16 | Stage II | 2 | 1 | 152.1 | 218.0 | 62 | tumor-free |
| ZLR-16 | 16/58 | Stage II | 2 | 1 | 73.0 | 235.6 | 92 | tumor-free |

Histologic subtype of all 16 samples is squamous cell carcinoma (SCC); NA indicates data are not available. Expression levels of E6/E7 are quantified by transcript per million (TPM) in RNA-seq data.

For most tumors, the number of breakpoints detected per sample was from 2 to 4, while 13, 12, and 10 breakpoints were detected in samples ZLR-06, ZLR-08, and ZLR-11, respectively.

Globally, integration breakpoints were scattered throughout the human and viral genomes, but there were some breakpoints in local hotspots (Fig. 1a). Integration breakpoints occurred throughout almost the whole virus genome. However, compared with predicted random distributions of breakpoints, they occurred more often in the *E1* gene ($P = 0.025$) and *E2* gene ($P = 0.009$) (Fig. 1b). These findings confirmed that in HPV integration, the disruption of the *E1* or *E2* gene (a negative regulator of oncogenes *E6* and *E7*) is preferred, which may lead to the dysregulation of oncoproteins E6 and E7, thereby promoting cervical carcinogenesis. In the host genome, integration breakpoints occurred more frequently in intragenic regions (43, 68.3%) than intergenic regions (20, 31.7%) ($P = 1.0e-4$) (Supplementary Table 2). In addition, integration breakpoints that occurred in the same sample tended to be clustered (Fig. 1c). For example, 12 breakpoints were clustered around the genomic region encompassing *LENG8*, *LENG9,* and *CDC42EP5* on chromosome 19 in sample ZLR-08, seven breakpoints were clustered around *CASC8* on chromosome 8 in sample ZLR-11 and 4 breakpoints were clustered around *HNF1B* on chromosome 17 in sample ZLR-12. These results suggested that HPV integration may induce local genomic instability, which then makes this affected region more easily integrated by other segmental HPV DNAs.

**Characterization of clonal integration events**. By taking advantage of long-read sequencing, we detected clonal integration events which were defined as a collection of chimeric reads (> 5 kb) containing at least one fragment of HPV DNA wrapped on both sides by the human genome (Table 1 and Supplementary Table 3). Nanopore technology is an amplification-free sequencing method, which allows direct and long-read sequencing of native DNA. Each chimeric read likely represents a single cell within the sequenced bulk cells. In 12 of 15 samples detected with integration breakpoints, at least one clonal integration event was identified with 3 of the 12 being particularly notable as having three or more clonal integration events identified. All clonal integration events with less than three supporting chimeric reads were further experimentally validated by PCR and Sanger

sequencing of their involving breakpoints (Supplementary Table 4 and Supplementary Fig. 1). In most tumors, one event was identified for each sample, corresponding to 2–3 breakpoints. The maximum number of events was observed in sample ZLR-08 where 12 breakpoints and 5 clonal integration events were detected. This was closely followed by ZLR-11 (10 breakpoints and 4 integration events) and ZLR-12 (4 breakpoints and 3 integration events).

The integration of high-risk HPV DNA into the host genome is a key molecular event in cervical carcinogenesis. After analyzing clonal integration events on a case-by-case basis, we identified various types of integrated HPV DNA segments. These integrated HPV DNA segments were frequently present in clonal integration events in one of four types. Type A represented a truncated HPV genome containing *E6/E7* genes (Fig. 2a). Type B was a truncated HPV genome that lacks *E6/E7* genes (Fig. 2b). Interestingly, compared with Type A, Type B didn't contain oncogenes *E6/E7* that are thought to be major contributors to cervical carcinogenesis. Type C was an overflowing continuous segment containing the intact HPV genome (Fig. 2c). In contrast with Type A and B, Type C incorporated at least one complete HPV genome and part of the HPV genome, in which *E1* and *E2* genes also appeared to be intact in some cases. Type D involved combinations of Type A, Type B or Type C (Fig. 2d), such as in the clonal integration event in tumor ZLR-02 containing two discontinuous viral genomes (i.e., Type A) being separated by 5 bp host sequences (left panel in Fig. 2d), or the clonal integration event identified in tumor ZLR-12 which was a combination of tandem amplification of Type A and one Type B with small host sequences linked between them (right panel in Fig. 2d). These four types of integrated HPV DNA segments appeared to be evenly presented in these clonal integration events detected in our samples. Specifically, the integrated HPV DNA segments of seven events were classified into Type A, five as Type B, six as Type C, and four as Type D (Fig. 3). Multiple types of integrated HPV DNA segments were frequently presented in a single tumor, suggesting the diversity and complexity of HPV integration in cervical cancer.

Based on the integration features identified above, we explored their association with clinical consequences (Table 1). To date, two patients have died of local relapse or distant metastases of cervical cancer. One patient was completely out of touch with us

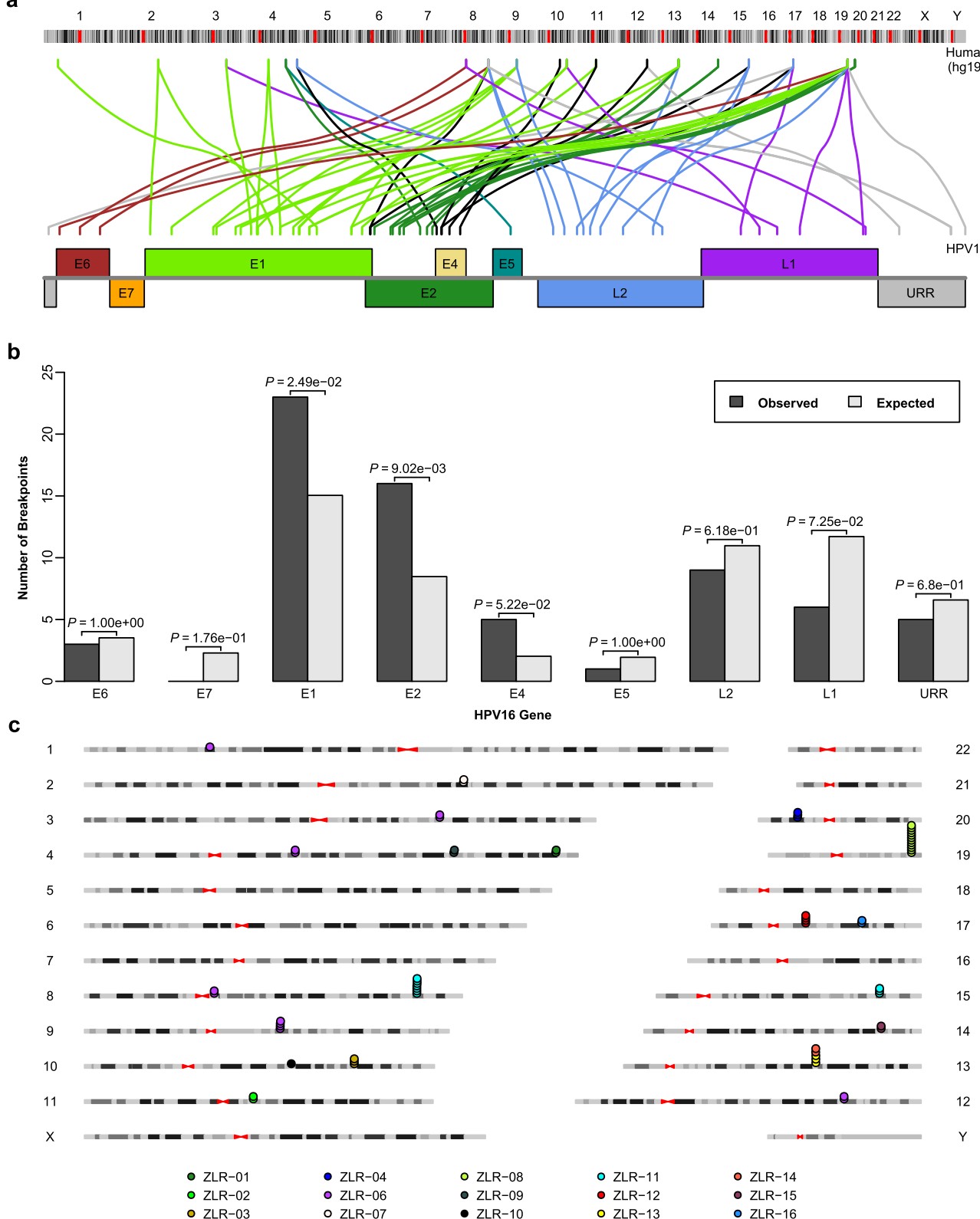

**Fig. 1 Integration breakpoints detected in the human and HPV genomes. a** Distribution of integration breakpoints across human and virus genomes. The colored link line indicates which human chromosome and which virus gene the integration breakpoint had occurred on. Only HPV16-related breakpoints are shown. **b** Comparison of the observed and expected numbers of breakpoints in the viral genome. The expected number of breakpoints was calculated based on the assumption that breakpoints are uniformly distributed across the viral genome. Only HPV16-related breakpoints are shown ($n = 61$). $P$ values were calculated by a two-sided binomial test. **c** Integration breakpoints are often clustered within the same sample in the human genome. Dots represent integration breakpoints and color keys indicate sample sources. Source data are provided as a Source Data file.

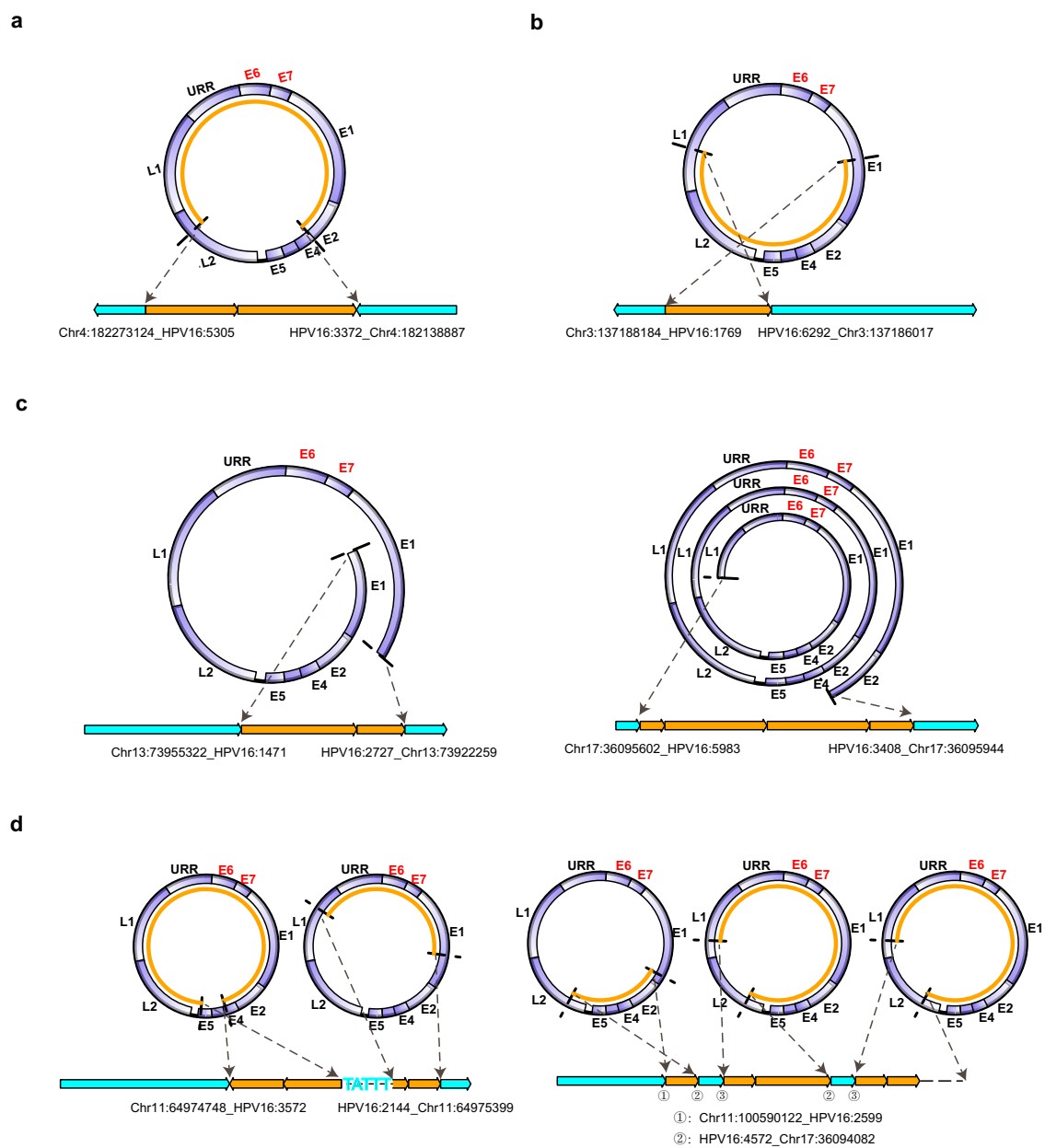

**Fig. 2 Four types of integrated HPV DNA segments in clonal integration events. a** Type A, a truncated HPV genome containing E6/E7. **b** Type B, a truncated HPV genome lacking E6/E7. **c** Type C, an overflowing continuous segment containing the intact HPV genome. **d** Type D, a combination of Type A, Type B, or Type C. In each panel, the two dashed arrows on the top indicate the HPV DNA fragment that was integrated into the human genome; the light blue and orange boxes on the bottom represent the human genome and virus genome, respectively. Source data are provided as a Source Data file.

and her status was unavailable. Thus, based on a dataset of 15 patients, a significant association between multiple integration events and poor prognosis (decease) was observed (Fisher's Exact Test; $P = 0.029$). While we did not find evident associations between other integration features (such as integration types) and clinical consequences.

**Breakpoints shared in clonal integration events**. Next, we analyzed the relationship between integration breakpoints and clonal integration events (Fig. 4 and Supplementary Table 3). As described above, the sample ZLR-08 was detected with 5 integration events and 12 breakpoints. Unexpectedly, only 6 out of 12 breakpoints were involved in these 5 integration events with 1

breakpoint (chr19:54983028_HPV16:2276) being commonly used in all 5 integration events (Fig. 4a). Similar results were also observed in samples ZLR-11 and ZLR-12. Specifically, in sample ZLR-11 the breakpoints chr8:128419658-HPV16:7784 and chr8:128323084-HPV16:4368 were the most common sites involved in 3 out of 4 events, and the other three sites chr15:85988611-HPV16:4774, chr15:86031825-HPV16:2797, and chr8:128323315-HPV16:478 were involved in 2, 1 and 1 events, respectively (Fig. 4b). In sample ZLR-12, 3 out of 4 detected breakpoints were involved in 3 clonal integration events, and these 3 breakpoints were involved in 3, 2, and 1 event, respectively (Fig. 4c and Supplementary Table 3). These findings indicate that some breakpoints are commonly used sites among

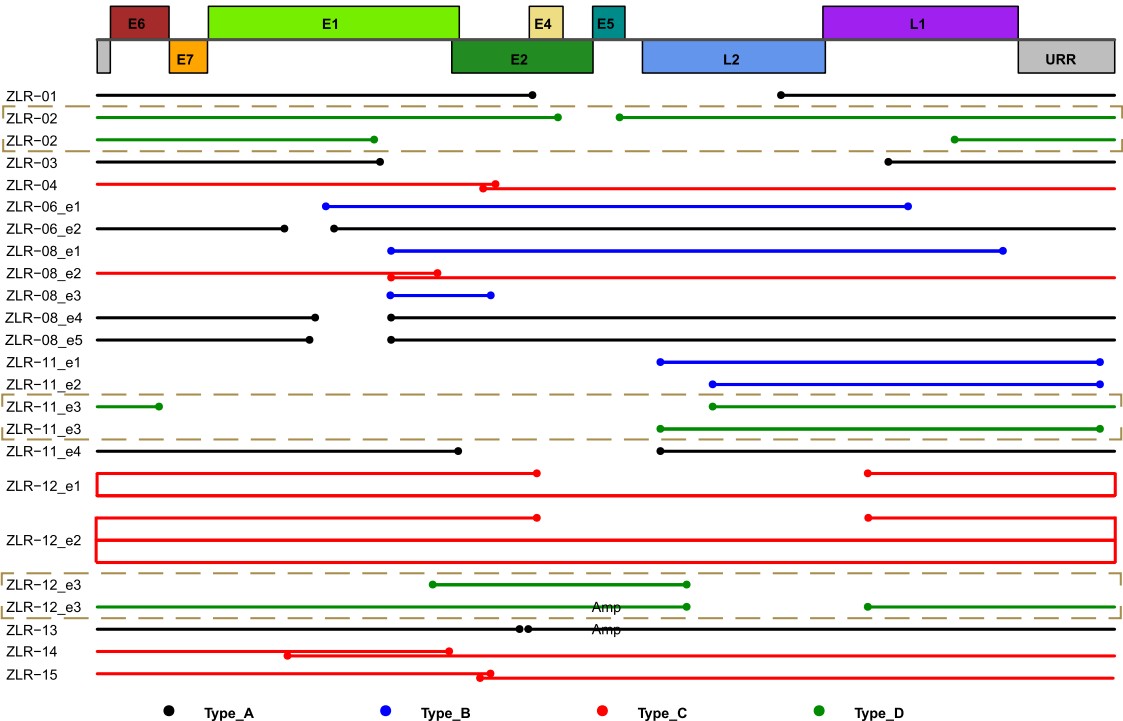

**Fig. 3 Schematic representations of integrated HPV DNA fragment in each clonal integration event.** The top rectangles represent a schematic diagram of the linearized HPV16 genome structure. The solid lines below the virus genome represent the DNA fragment of HPV that was integrated into the human genome in each clonal integration event. The dot at the line end represents the breakpoint in the viral genome. The line/dot color represents a respective type of integrated HPV DNA segments as follows: Black, Type A; Blue, Type B; Red, Type C; and Green, Type D. The note of "Amp" on the lines indicates that these DNA fragments are amplified in the clonal integration event. If there was more than one event in a sample, a designation "en" is added after the sample name to distinguish these events, where n codes 1 to number of events. Only HPV16-related clonal integration events are shown. Source data are provided as a Source Data file.

clonal integration events, and these commonly used breakpoints likely originated from an ancestral clonal integration event, in which the first HPV integration occurred in cervical epithelial cells.

Based on the shared breakpoints discovered above, we further analyzed their possible intrinsic links to intratumor heterogeneity in cervical cancer. Figure 4c is a schematic representation of three captured authentic chimeric reads that represent three different integration events in ZLR-12. The junction site ① was the shared site of three distinct integration events and its left stretched sequence in these integration events was captured by long-read sequencing. Since both the normal flanking human chr17 sequence (in e1 and e2) and the truncated and stitched hybrid sequence (in e3) were present, these three distinct integration events are unlikely to occur in the same cell due to the diploid nature of human cells. Furthermore, the sequencing depth reported by sniffles for junction sites ①, ②, and ④ were 58, 44, and 18, respectively. This suggested that the junction site ④ occurred less frequently among these junction sites, even though it was tandemly duplicated in ZLR-12_e3. A plausible explanation is that (i) the shared breakpoint ① may originate from the ancestral clonal integration event, in which the first viral integration occurred; (ii) these integration events sharing the common breakpoint ① are likely evolutionarily related; and (iii) the integration event ZLR-12_e3 may be a later event during cervical cancer progression. These observations provide evidence that these integration events may be involved in clonal evolution, thereby contributing to another layer of intratumor heterogeneity in cervical cancer in addition to somatic mutations and copy-number alterations.

In addition to ZLR-12, two other patients (ZLR-08 and ZLR-11) carrying shared breakpoints were also inferred to bear some intratumor heterogeneity. Patient ZLR-8 completely lost contact with us and her status was unavailable, while patients ZLR-11 and ZLR-12 died of tumor recurrence, suggesting that intratumor heterogeneity may contribute to their poor prognosis (Fisher's Exact Test; $P = 0.009$).

**Integrated HPV DNA-mediated translocation between chromosomes.** In addition to shared breakpoints, some clonal integration events were involved in inter-chromosomal translocations (Table 2). In these, two sides of the virus genome were respectively fused with different chromosomes based on long chimeric reads. For example, three clonal integration events identified in sample ZLR-11 were related to translocations between chromosomes 8 and 15 (Fig. 4b); one clonal integration event identified in sample ZLR-12 was related to translocations between chromosomes 11 and 17 (Fig. 4c). Several previous studies based on Sanger sequencing or short-read NGS technologies have implied that HPV-associated chromosomal translocations may exist[25,37]. Here, taking advantage of long-read sequencing, we directly identified the authentic and relatively complete chromosomal translocations mediated by integrated HPV DNA. These findings suggest that HPV integration not only disrupts genes around integration breakpoints, but also may trigger greater genomic instability involving chromosomal translocations.

**ECC virus-human DNA structures.** Furthermore, seven clonal integration events were potentially involved in the formation of ECC DNA structures in four samples (Table 2). Among these

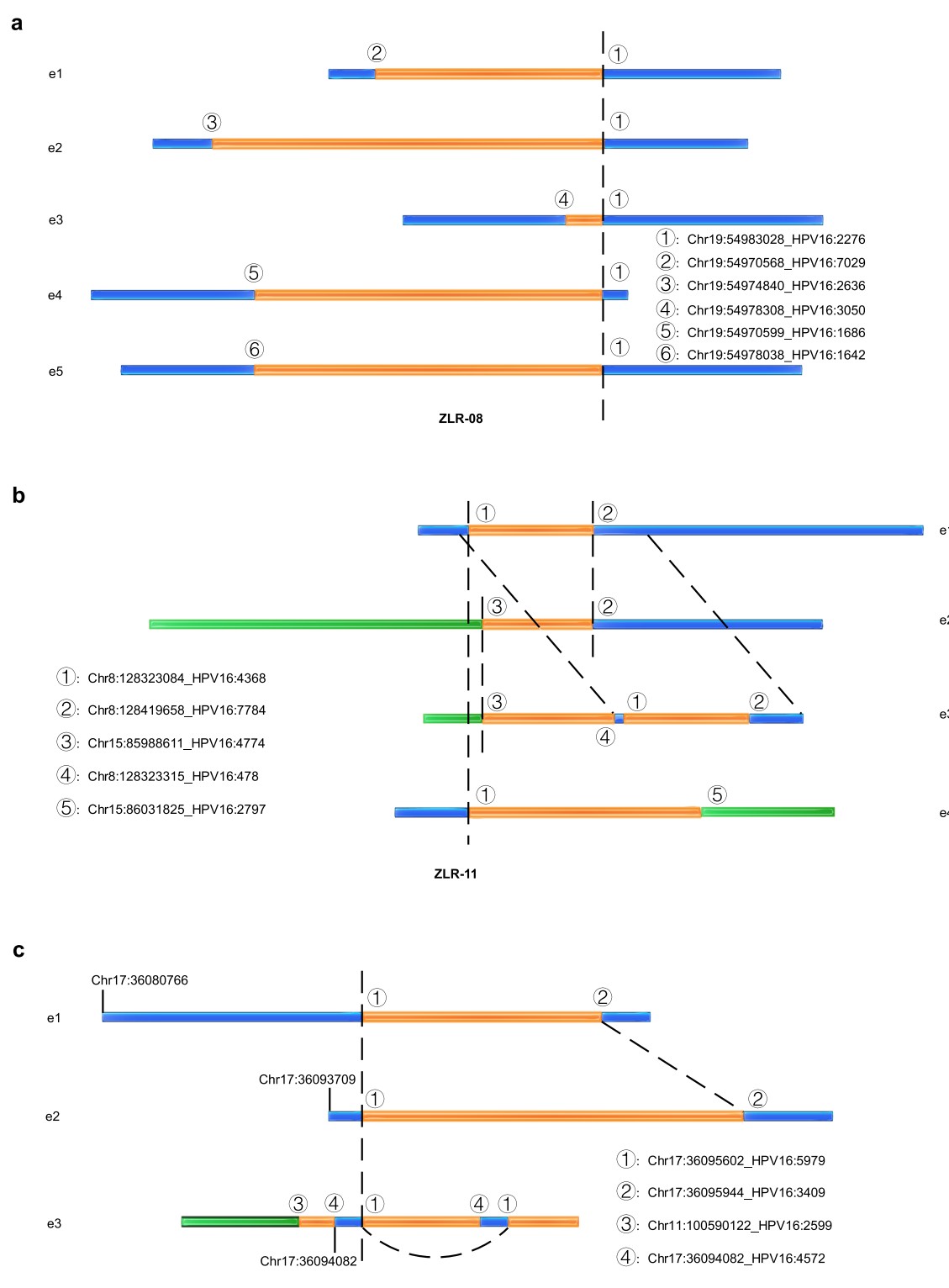

**Fig. 4 Breakpoints shared in clonal integration events and HPV integration-related inter-chromosomal translocations. a–c** Schematic illustrations of authentic chimeric reads representing clonal integration events that participated in the use of shared breakpoints: **a** ZLR-08, **b** ZLR-11, and **c** ZLR-12. The orange box represents the virus genome; the blue and green boxes represent different chromosomes of the human genome. Identical breakpoints were designated by the same number and were linked by dashed lines. Clonal integration events in ZLR-11 and ZLR-12 induced translocations between chromosomes. Source data are provided as a Source Data file.

**Table 2 Human genes potentially disrupted in HPV clonal integration events[*].**

| Chr | Start | End | Distance | Type | Region | Genes | Consequences | Events[*] |
|---|---|---|---|---|---|---|---|---|
| 4 | 182138890 | 182273142 | 134251 | DEL | intergenic | LINC00290;TEMN3-AS1 | dist = 58588;dist = 468016 | ZLR-01 |
| 11 | 64974748 | 64975398 | 649 | DEL | intronic | CAPN1 | NA | ZLR-02 |
| 10 | 104080707 | 104084259 | 3551 | DEL | intronic | GBF1 | NA | ZLR-03 |
| 20 | 14754684 | 14840951 | 86266 | DEL | intronic | MACROD2 | NA | ZLR-04 |
| 3 | 137186017 | 137188183 | 2165 | UD | intergenic | IL20RB;SOX14 | dist = 456091;dist = 294951 | ZLR-06_e1 |
| 9 | 75413449 | 75532466 | 119016 | DEL/ECC | exonic | ALDH1A1;TMC1;LINC01474 | frameshift deletion | ZLR-06_e2 |
| 19 | 54970568 | 54983028 | 12460 | DEL/ECC | exonic | CDC42EP5;LENG8;LENG9 | frameshift deletion | ZLR-08_e1 |
| 19 | 54974840 | 54983028 | 8188 | DEL/ECC | exonic | CDC42EP5;LENG9 | frameshift deletion | ZLR-08_e2 |
| 19 | 54978308 | 54983028 | 4720 | UD | UTR5 | CDC42EP5 | NA | ZLR-08_e3 |
| 19 | 54970599 | 54983028 | 12429 | DEL/ECC | exonic | CDC42EP5;LENG8;LENG9 | frameshift deletion | ZLR-08_e4 |
| 19 | 54978038 | 54983028 | 4990 | DEL/ECC | UTR5 | CDC42EP5 | NA | ZLR-08_e5 |
| 8 | 128323084 | 128419658 | 96573 | DEL/ECC | ncRNA_exonic | CASC21;CCAT2;CASC8 | NA | ZLR-11_e1 |
| 15 | 85988611 | 85988611 | 0 | TRA | intronic | AKAP13 | NA | ZLR-11_e2 |
| 8 | 128419658 | 128419658 | 0 | TRA | ncRNA_intronic | CASC8 | NA | ZLR-11_e2 |
| 15 | 85988611 | 85988611 | 0 | TRA | intronic | AKAP13 | NA | ZLR-11_e3 |
| 8 | 128323315 | 128323315 | 0 | TRA | ncRNA_intronic | CASC21;CASC8 | NA | ZLR-11_e3 |
| 8 | 128323084 | 128323084 | 0 | TRA | ncRNA_intronic | CASC21;CASC8 | NA | ZLR-11_e3 |
| 8 | 128419658 | 128419658 | 0 | TRA | ncRNA_intronic | CASC8 | NA | ZLR-11_e3 |
| 15 | 86031825 | 86031825 | 0 | TRA | intronic | AKAP13 | NA | ZLR-11_e4 |
| 8 | 128323084 | 128323084 | 0 | TRA | ncRNA_intronic | CASC21;CASC8 | NA | ZLR-11_e4 |
| 17 | 36095602 | 36095944 | 341 | DEL | intronic | HNF1B | NA | ZLR-12_e1/e2 |
| 11 | 100590122 | 100590122 | 0 | TRA | intronic | ARHGAP42 | NA | ZLR-12_e3 |
| 17 | 36094082 | 36095602 | 1519 | AMP | intronic | HNF1B | NA | ZLR-12_e3 |
| 13 | 73982892 | 73982911 | 18 | DEL | intergenic | KLF5;LINC00392 | dist = 331213;dist = 155426 | ZLR-13 |
| 13 | 73982911 | 73989111 | 6199 | AMP | intergenic | KLF5;LINC00392 | dist = 331231;dist = 149270 | ZLR-13 |
| 13 | 73980114 | 73982892 | 2777 | AMP | intergenic | KLF5;LINC00392 | dist = 328434;dist = 155489 | ZLR-13 |
| 13 | 73922258 | 73955322 | 33063 | DEL/ECC | intergenic | KLF5;LINC00392 | dist = 270578;dist = 183059 | ZLR-14 |
| 14 | 91451918 | 91451921 | 2 | DEL | intronic | RPS6KA5 | NA | ZLR-15 |
| 17 | 57757805 | 57879164 | 121359 | DEL | exonic | CLTC;PTRH2;VMP1 | frameshift deletion | ZLR-16 |

[*]e integration events, AMP amplification, DEL deletion, ECC extrachromosomal circular DNA inferred from chimeric reads, TRA translocation, UD undetermined.

events, the cellular sequences on both sides of the integrated HPV DNA fragment were reversed. Elevated base coverages of involved regions were also found in 6 out of 7 events, especially compared with decreased base coverages commonly observed in other deletion types of events (Supplementary Fig. 2). Specifically, as shown in the chimeric reads of sample ZLR-11 (event 1), the host-derived reads on both sides of the HPV DNA fragment were reversed end to end in the human genome (Fig. 5a). Two potential mechanisms of this structure can be speculated. In the first, both flanking human sequences undergo breakage and inversion, resulting in two chromosomal rearrangements interspersed with viral sequences. In the second, the two ends of the integrated HPV DNA fragment are connected with the human sequences on both sides to form an ECC virus-human hybrid DNA structure (Fig. 5b). Additionally, the involved region on the host genome was focally amplified according to the DNA base coverage (Fig. 5a), which was subsequently verified by PCR (Fig. 5c), suggesting the existence of ECC virus-human hybrid and its autonomous replication. To further confirm the existence of ECC hybrid structure (ZLR-11_e1), we removed the linear DNA from the sample ZLR-11 using an exonuclease treatment for 5 days with a linear gene COX5B as a negative control and three internal plasmids as positive controls. As a result, the two virus-human junction sites could still be successfully detected

after the removal of linear DNA (Supplementary Fig. 3). These findings consistently support the existence of the circular DNA structure of the ZLR-11_e1 integration event.

**Genes disrupted by HPV integration.** We then identified genes potentially disrupted by HPV integration, including those directly disrupted by HPV integration or those closest to HPV integration sites. As a result, a total of 34 genes (including 27 protein-coding genes and 7 non-coding RNAs) were annotated near integration breakpoints (Supplementary Table 2). The intergenic region between genes KLF5 and LINC00392 on chromosome 13 was a recurrent hotspot integration site that harbored 5 breakpoints across two samples. Other genes at previously reported hotspots of HPV integration included CASC8, CASC21, MACROD2, TEX41, and VMP1 (alias TMEM49)[38–40]. 

Owing to the advantage of long-read sequencing, besides the single site of breakpoints, we also annotated the affected regions on the human genome resulting from HPV integration based on the identified complete clonal integration events (Table 2). Among 22 clonal integrated events, the partial deletion of human genome sequences was the most common form, and the length of deletion fragments ranged from 2 to 134,251 bp. By contrast, as shown in the cases of ZLR-12 (event 3) and ZLR-13, some fragments of the human genome were co-amplified with the

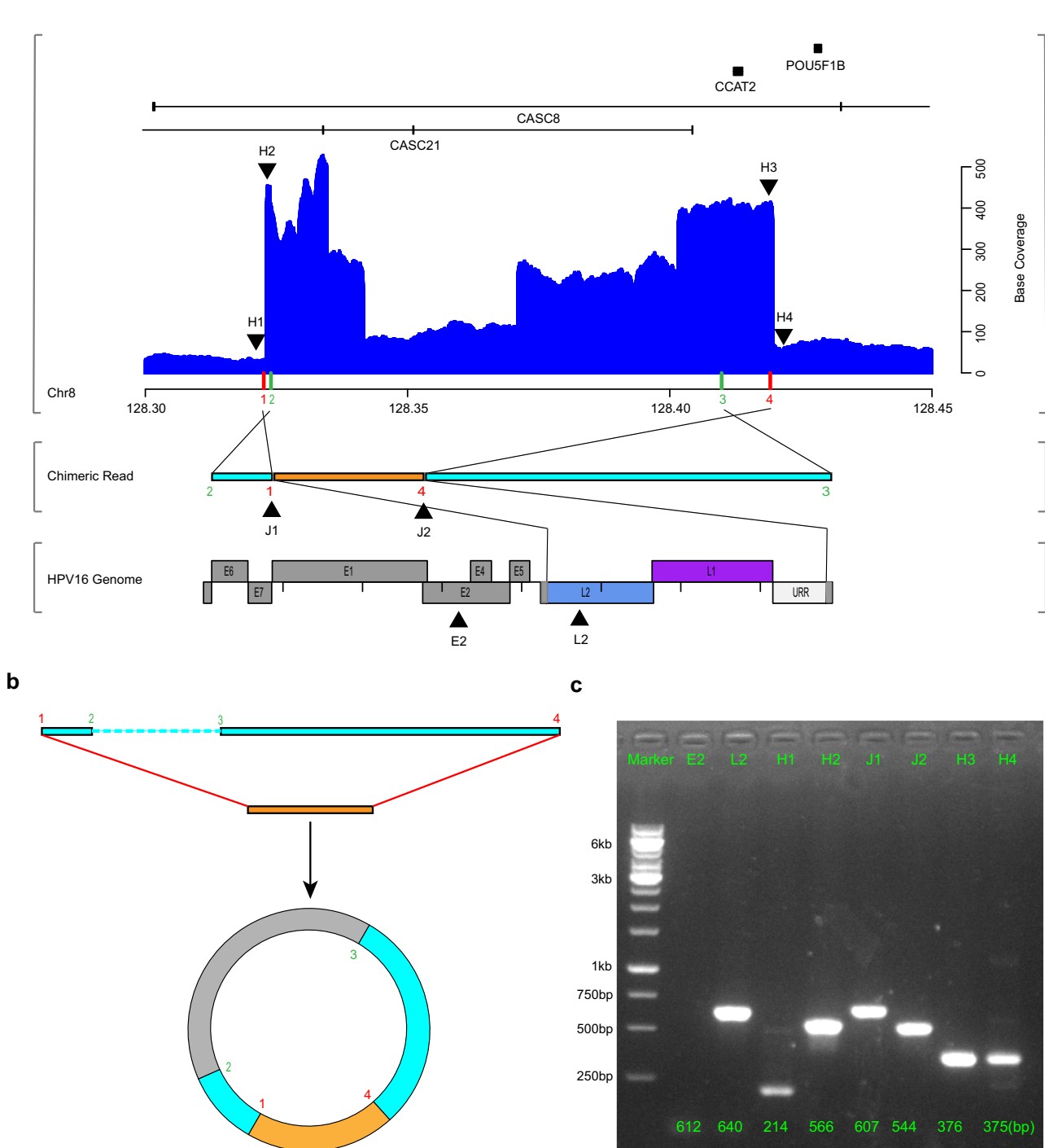

integrated HPV DNAs. Another common form was inter-chromosomal translocations mediated by the integrated HPV DNAs, such as the clonal integration event in sample ZLR-11. In addition, integration-related ECC DNA that often led to partial deletion and focal amplification of cellular sequences was also found to be considerably prevalent in these clonal integration events. As a result, a total of 25 genes were identified in these 22 affected regions. It is worth noting that the integrated HPV DNA segments of five clonal integrated events belonged to Type B which lacks oncogenes *E6* and *E7* (Fig. 3). Genes potentially affected by these Type B events include *IL20RB*, *SOX14*, *LENG8*,

*LENG9*, *CDC42EP5*, *CASC21*, *CCAT2*, *CASC8,* and *AKAP13*. Enrichment analysis of gene ontology (GO) terms and KEGG pathways showed that these genes potentially disrupted by HPV integration were involved in Rho protein signal transduction and in the regulation of cytokine production (Supplementary Fig. 4).

Combined with the corresponding RNA-seq data (Supplementary Table 5), we then further explored the impact of HPV integration on the expression of these genes around breakpoints or on the affected genomic regions related to the clonal integration events. Compared with those without corresponding HPV integration, approximately 23.7% of such genes were

**Fig. 5 Extrachromosomal circular virus-human DNA structures. a** A chimeric read (middle) from ZLR-11_e1 was mapped to both human (top) and HPV16 (bottom) genomes. In top panel, the blue histogram shows base coverage of regions of the human genome around HPV integration, with indicated chromosomal loci (down, x-axis in megabases) and corresponding gene schematics (up). On the x axis, red bars indicate the positions of HPV insertion breakpoints in the human genome and green bars indicate the positions of two endpoints of the observed chimeric read (middle). The colored Arabic numerals indicate the order of these breakpoints in the human genome, which indicates human-derived sequences on both sides of the chimeric read were in reversed orientations. The eight black triangles indicate the regions selected for PCR validation in **c** below. In bottom panel, the colored regions in the HPV16 genome represent the sequences contained in the integrated structure and gray regions represent the sequences that were replaced or lost due to integration. **b** Extrachromosomal circular virus-human DNA structure. The two ends of the integrated HPV DNA fragment (orange box) are connected with the human sequence on both sides (light blue box) to form an ECC DNA structure (not drawn to scale). The unobserved region (~86 kb) was indicated by gray in the circular structure. **c** PCR validation of amplification of eight selected regions in the clonal integration event ZLR-11_e1. Three independent experiments give similar results. IDs of the selected regions are indicated at the top. L2, H2, J1, J2, and H3 are five regions in the ECC DNA structure that are supposed to be amplified. E2, H1, and H4 are the control regions around the target amplification regions of HPV and human, respectively. The numbers at the bottom are the predicted PCR product sizes. Source data are provided as a Source Data file.

significantly altered in their expression levels upon HPV integration, including *LINC00290*, *LENG9*, *CCAT2*, *ARHGAP42*, *HNF1B*, *FOXD2*, *TMC1*, *IL15*, and *RPS6KA* (Supplementary Fig. 5). Most of these genes have been reported to be associated with cancer, while *LENG9* is rarely reported in the literature and has not been implicated in cancer, suggesting that *LENG9* may be a potential tumor driver gene in cervical cancer. *LENG9* was significantly upregulated upon HPV integration in sample ZLR-08. Its expression was highest among all 103 tumors we sequenced in this and previous studies[41], representing an ~10-fold increase in expression levels compared to the average expression level of these tumors (Fig. 6b; z score = 16.24, false discovery rate (FDR) = 1.21e-58). A total of 12 breakpoints were detected in ZLR-08, of which 6 were involved in 5 detected integration events, with 1 breakpoint being commonly used in all 5 integration events (Table 1 and Supplementary Table 3). In addition, all 12 breakpoints were clustered in 19q13.42. Accordingly, the genomic region associated with these 5 integration events contained 3 genes: *LENG8*, *LENG9,* and *CDC42EP5* (Table 2). Based on the DNA sequencing base coverage around this target region (Fig. 6a), two significant focal amplifications were observed, corresponding to sequence fragments in ZLR-08_e1/e4 and ZLR-08_e2, respectively (Table 2). Interestingly, all these three integration events could potentially form virus-human hybrid ECC structure that can replicate autonomously and lead to amplification. The other two events ZLR-08_e3 and ZLR-08_e5 occurred less frequently according to the sequencing depth of their involving breakpoints (Supplementary Table 3), which may be an explanation for the absent manifestation of these two events on the DNA depth plot (Fig. 6a). In particular, it was observed that only the gene *LENG9* was completely amplified in this target region, resulting in its highest expression among all 103 cervical tumors (Fig. 6b).

To evaluate the potential functional role of *LENG9* in cervical carcinoma, we overexpressed *LENG9* in cervical cancer cell lines CaSki and SiHa (Fig. 6c), as *LENG9* expression was significantly upregulated in HPV-integrated cervical tumors (i.e., ZLR-08). For the overexpression assay, a full-length human *LENG9* sequence was obtained by PCR and subcloned into the PLVX-puro vector by Lipofectamine 3000 to establish cells that stably overexpressed *LENG9*. Consistently, upregulation of *LENG9* in CaSki and SiHa cells significantly increased proliferation, migration, and invasion of cervical cancer cells (Fig. 6d–f). Furthermore, we performed knockdown assays by treating CaSki cells with siRNAs targeting *LENG9*. As a result, depletion of LENG9 significantly inhibited cell proliferation, migration, and invasion of cervical cancer cells (Supplementary Fig. 6). Taken together, these results suggest that LENG9 may play an oncogenic role in tumor pathogenesis by promoting tumor cell proliferation, migration, and invasion.

In addition, we particularly focused on one HPV-integrated tumor (i.e., ZLR-01) with relatively low expression of *E6* and *E7* (Table 1 and Supplementary Fig. 7) whose dysregulation had been considered to be primarily responsible for cervical carcinogenesis. This suggested that alternative oncogenic pathways independent of E6/E7 activation may be occurring for this tumor. Disruption of cancer-related genes on the host genome by the action of HPV integration per se is the most likely explanation. In the ZLR-01 sample, only two breakpoints and one single associated clonal integration event were detected. The integration event led to a 134,251 bp (chr4:182138890–182273142) loss of 4q34.3 region encompassing a long intergenic non-coding RNA (lincRNA) gene *LINC02500*. Another lincRNA gene *LINC00290*, located approximately 58.6 kb downstream of the breakpoint, was found to be expressed only in this tumor, but with no corresponding expression in the other tumors investigated (Supplementary Fig. 7). Although the biological function of these two lincRNA genes is yet to be determined, several previous studies have suggested that they are of cancer relevance. A pan-cancer analysis (including 11 cancer types without cervical cancer) of somatic copy-number alteration revealed that the genomic region comprising these two genes was a recurrent focal deletion region across multiple cancer type[42]. Similarly, the focal depletion region around *LINC00290* and *LINC02500* was also frequently observed in *TP53*-mutated childhood adrenocortical tumors[43]. It has been also reported that a sharper region encompassing only *LINC02500* gene was a significant target loss region in endometrial carcinoma[44]. Possibly due to the lack of annotation for *LINC02500* and low-resolution copy-number analysis, previous studies usually picked *LINC00290* in this focal deletion region as a tumor suppressor candidate. However, our current large-range high-resolution results suggest the recently annotated *LINC02500* may be a tumor suppressor candidate gene, while *LINC00290* may be an oncogene candidate gene. These candidate genes remain to be further studied.

**Characterization of SVs.** Finally, we evaluated the potential biological relevance of somatic SVs detected by long-read sequencing to HPV integration. We first checked the overall somatic SV landscape in cervical cancer with HPV integration. The median number of purified somatic SVs per genome was 5324. However, an extremely high SV burden was found in ZLR-08 tumor. INVDUP and DEL were the main SV types, accounting for 78.0% of the total SVs (Supplementary Fig. 8a, b). In four of five SV types, the SV length was around 1 kb or less, while a stretched distribution of INV length was observed (Supplementary Fig. 8c).

Comparative analysis of SV from regions with or without HPV integration can uncover their association (Supplementary Table 6). To control the genomic background, we mainly focused

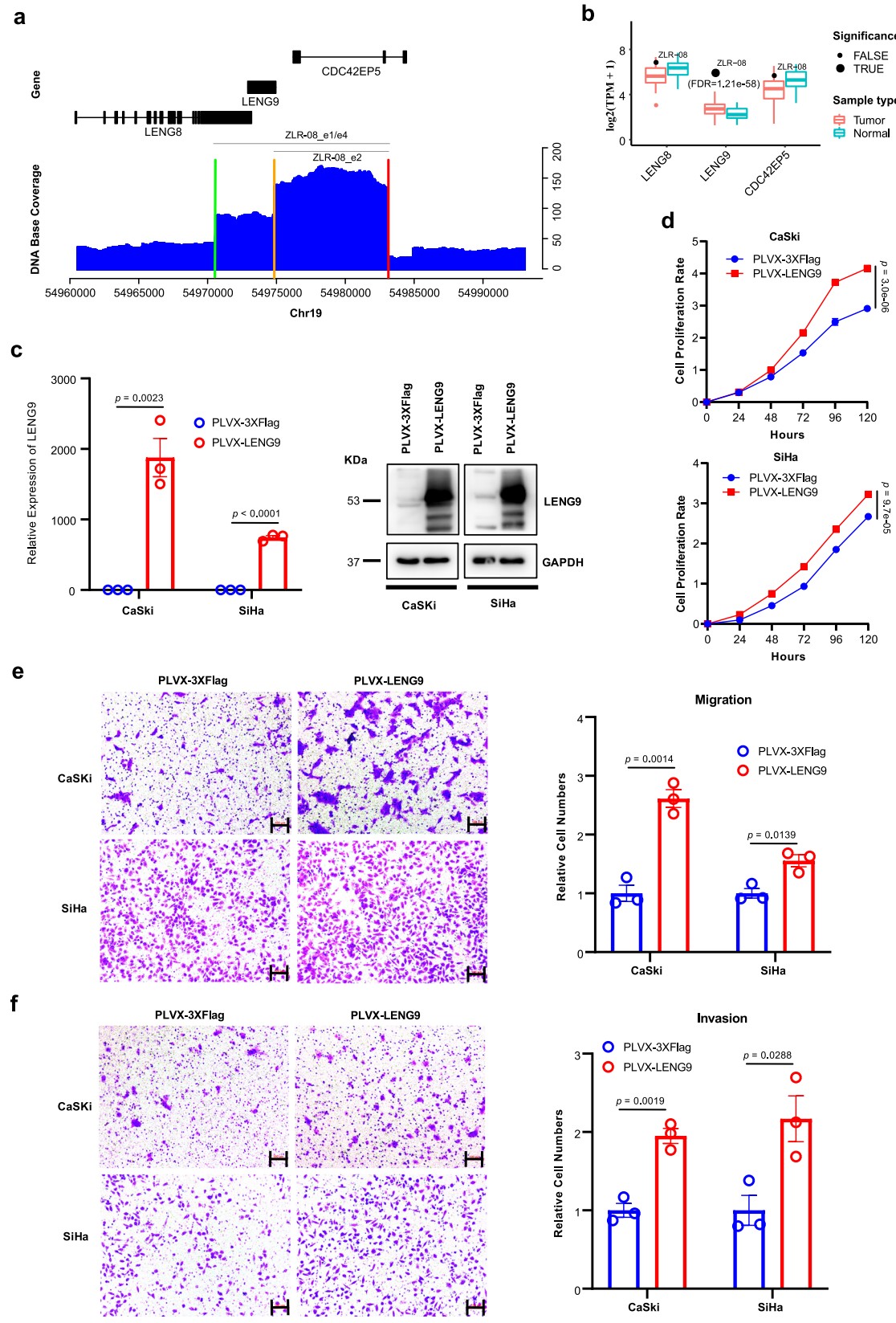

on genome windows spanning HPV integration sites where a series of window sizes were selected. According to the sample source of data in the corresponding window, if the window contained an exact HPV integration event, the region was grouped as an intSample, otherwise, it was grouped as an intRegion. An elevated SV count was observed in intSample compared to intRegion in a series of windows from 1 kb to 500 kb (Fig. 7a). Among the five SV types, DUP was enriched near

**Fig. 6 Functional analysis of LENG9 in cervical cancer cell lines. a** DNA sequencing coverage depth around the target region involved in HPV integration. The blue histogram shows base coverage (y-axis) with indicated chromosomal loci (x axis) and corresponding gene schematics (top). Colored bars on the x axis indicate the positions of HPV insertion breakpoints in the human genome. Gray segments indicate the human genomic segments involved in integration events (target region). **b** Expression levels of three genes located in the target region in all tumors ($n = 103$) and adjacent normal samples ($n = 39$). Data are shown in boxplots. The thick line in the box is median and the box spans from Q1 (25th percentile) to Q3 (75th percentile). The whiskers extend to the most extreme observation within 1.5 times the interquartile range (IQR = Q3–Q1) from the nearest quartile. The gene expression level in ZLR-08 is highlighted by black dots. Significantly upregulated genes with adjusted p value < 0.05 are highlighted by dots of larger size. **c** The expression level of LENG9 was determined by qRT-PCR and western blot in cervical cancer lines CaSki and SiHa transfected with LENG9 overexpression or mock vector. Data are presented as mean values ± SEM ($n = 3$). **d–f** Proliferation, migration, and invasion assays of CaSki and SiHa transfected with LENG9 overexpression or mock vector. Scale bar, 100 μm. Quantification of migration and invasion cells was summarized as histograms. Data are presented as mean values ± SEM ($n = 5$ for proliferation assays and $n = 3$ for migration and invasion assays). Two sample t test was used for comparing the difference between overexpression and control groups. Source data are provided as a Source Data file.

---

integration sites (Fig. 7b). DUPs near HPV integration sites had larger sizes than the other SV types (Fig. 7c), suggesting a greater genomic impact of HPV integration on DUP. To further investigate the spatial relationship between HPV integration and SVs, the distance distribution of SV to adjacent HPV integration sites was estimated. An increased proportion of SVs located close to HPV integration sites was observed in DUP and INVDUP. Among these SVs, the main peak was positioned almost at the HPV integration site (Fig. 7d).

Considering the enriched SV occurrence near HPV integration sites, we next compared five genomic features of SVs from different regions to gain information on the origin of HPV integration-related SVs. Several genomic features showed significant deviations of SVs in intSample from that in intRegion, such as those of replication timing, GC content, recombination value, and the distance to topologically associating domain (TAD) boundaries (Fig. 7e, f, and Supplementary Fig. 9). The strong association between early replication timing and DUP and INVDUP near HPV integration sites is shown in Fig. 7e, and is consistent with previous observations[45]. Early replication is known to be associated with high DNA double-strand break (DSB) regions[46], implying a potential preference for HPV integration-related SVs arising from DSBs. Disruption of TAD boundaries by structural variants has also been reported to cause developmental disorders[47]. In our data, the relative proximity of SVs and TAD also reveals the potential consequence of SVs occurring near HPV integration sites (Fig. 7f).

To further understand its role in carcinogenesis, the functional implications of HPV integration-related SVs were then estimated. More than half of the genes disrupted by DUP, INS, or INVDUP near HPV integration had strong evolutionary constraint, suggesting their essential structural and function integrities in general populations (Fig. 7g). To explore the impact of genomic alteration at the transcriptome level, we also checked the expression change of the SV-covered transcribable regions by using RNA-seq data from the same set of long-read sequencing samples, involving 9904 genes. For each disrupted gene, a Z score was used to estimate the expression change in samples with SVs, as compared to other samples. In this, 33.3% of genes had aberrantly high expressions in DUPs (Fig. 7h). Taken together, these results suggest that HPV integration induces genomic instability and SVs that have substantial functional consequences in cervical carcinogenesis.

## Discussion

Integration of high-risk HPV DNA into the human genome is considered an important molecular event in cervical carcinogenesis. However, current short-read NGS technologies have proven inadequate to provide a complete picture of the large-range HPV integration events and many SVs accompanying HPV integration. In this study, we performed a genome-wide analysis

of HPV integration in 16 HPV16-positive primary cervical tumors using Nanopore long-read sequencing and RNA sequencing technologies. Taking advantage of long-read sequencing, we were able to characterize at high-resolution the various structures of the integrated HPV DNA segments that are frequently present in clonal integration events. We further revealed the spatial relationship between HPV integration and various SVs as well as their functional consequences in cervical cancer. Combining this with the corresponding RNA-seq data, we were able to pinpoint LINC00290, LINC02500, and LENG9 as potential driver genes in cervical cancer.

Long-read sequencing analysis enabled us to sequence long chimeric reads that contain at least one fragment of HPV DNA wrapped on both sides of the human genome. This facilitates the identification of a number of clonal integration events and the revealing of several significant characteristics of HPV integration in cervical cancer. Firstly, in the host genome, integration breakpoints that occurred in the same sample tended to be clustered. Sample ZLR-11 was a good example where 7 of 10 breakpoints were clustered on 8q24.21 (Fig. 1c). A possible explanation for these intra-sample breakpoint clusters is that HPV integration induces local genomic instability, then making this affected region more easily integrated by other HPV DNA fragments. Secondly, shared breakpoints have been shown to be involved in multiple clonal integration events. For example, we observed that one of breakpoints was involved in all three of the clonal integration events identified in sample ZLR-12 (Supplementary Table 3). These shared breakpoints may originate from an ancestral clonal integration event, in which the first viral integration occurred in cervical epithelial cells. Thirdly, taking advantage of long-read sequencing technologies, our analysis directly revealed some authentic HPV-mediated inter-chromosome translocations. In these events, HPV acts as a linker to fuse with two different human chromosomes. This is best illustrated in sample ZLR-11. This finding suggests that HPV integration not only influences genes around integration breakpoints, but also can induce extensive genomic disruption through chromosomal translocation. Finally, it is generally believed that HPV integration is via the introduction of a viral genome sequence into the host genome while the present study demonstrates another potential form of integration where HPV acts as a scissor and the host genomic fragment is excised from the chromosome to form a virus-human hybrid ECC structure. ECC DNA is an important genomic feature in cancers and plays an important role in driving tumor evolution and genetic heterogeneity[48]. Our analysis demonstrated that 7 out of 22 clonal integration events were potentially involved in the formation of ECC virus-human DNA structures, which is much more prevalent than expected. Notably, several ECC DNAs belong to Type B events lacking oncogenes E6 and E7, but they were detected to have high base coverage, implying strong co-amplification of virus and human genomes.

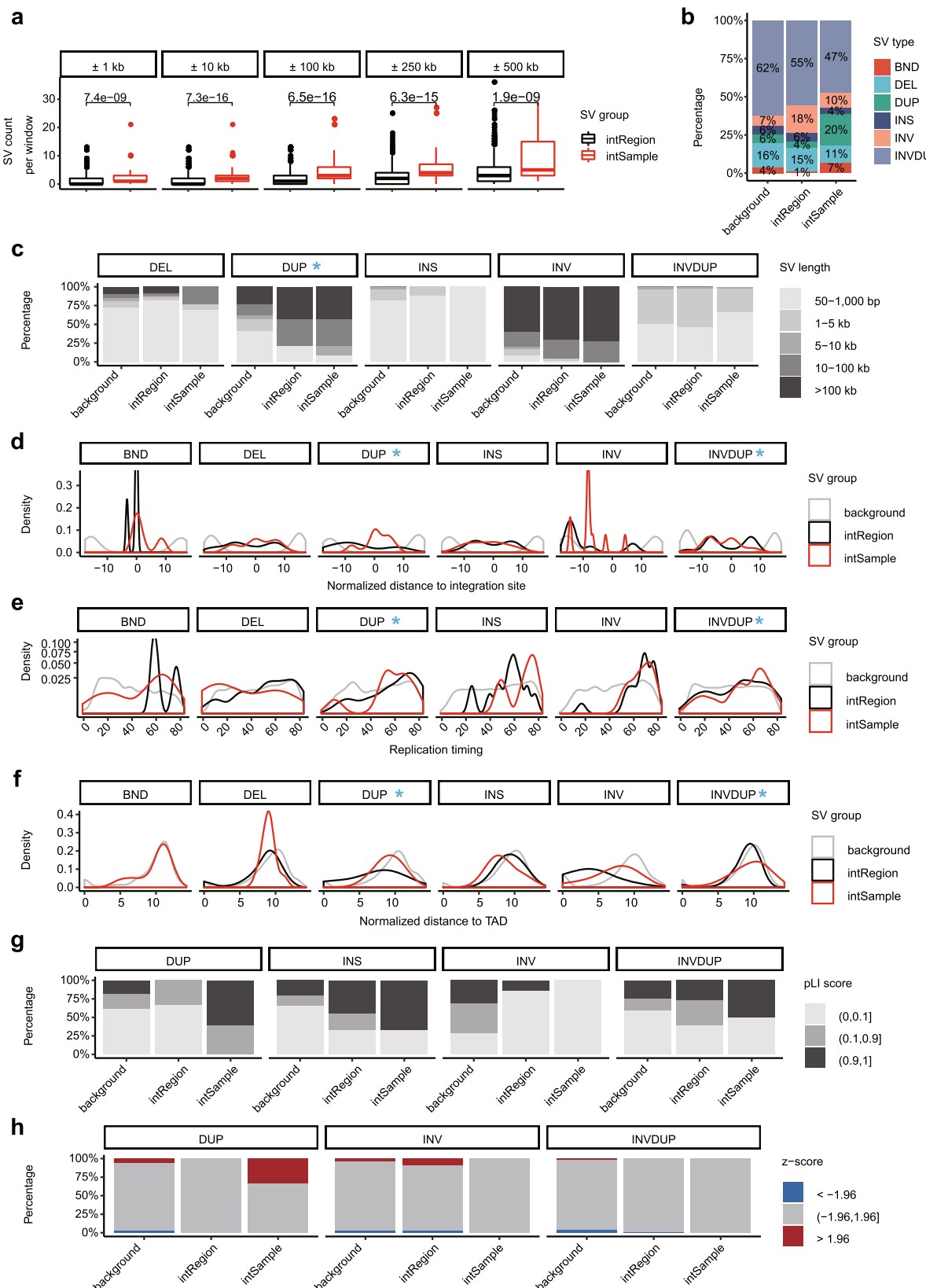

For example, such ECC DNA event led to the focal amplification of an oncogenic lncRNA *CCAT2* in tumor ZLR-11 (Fig. 5). *CCAT2* is upregulated in multiple types of cancer and plays an oncogenic role in cell proliferation, invasion, and metastasis[49]. These results suggest that virus-induced ECC DNA may be a common alternative mechanism of viral carcinogenesis in cervical cancer that warrants further investigation.

Our study reveals that these integration events may contribute to another layer of intratumor heterogeneity in cervical cancer in addition to somatic mutations and copy-number alterations. Of

**Fig. 7 The relationship between structural variations and HPV integration. a** SV counts in windows of different lengths compared between intRegion ($n = 1232$) and intSample ($n = 88$). Data are shown in boxplots. The thick line in the box is median and box spans from Q1 (25th percentile) to Q3 (75th percentile). The whiskers extend to the most extreme observation within 1.5 times the interquartile range (Q3–Q1) from the nearest quartile. **b** The relative proportion of SV types in different regions. **c** The SV size distribution of each SV type in different regions. **d** Distribution of normalized distance from SVs to nearest HPV integration sites. **e** Distribution of SV replication timing of each SV type in different regions. **f** Distribution of normalized distance from SVs to nearest TAD. **g** The distribution of SV pLI score in different regions for each SV type. **h** The composition of SV expression changes in different regions for each SV type. BND, breakend; DEL, deletion; DUP, duplication; INS, insertion; INV, inversion; INVDUP, inverted duplication. In each genomic feature, the SV types that have distinct distribution deviations between intSample and intRegion are highlighted with a blue arrow or asterisk. For each feature in every displayed SV type, the statistical significance of the difference between intSample and intRegion was summarized in Supplementary Table 6. Source data are provided as a Source Data file.

note, a recent study identified one of eight tumors displaying intratumor heterogeneity with respect to junction distribution through microdissection of tumor tissues[50]. In the present study, the potential existence of intratumor heterogeneity of HPV integration in cervical cancer was inferred based on the following observations: First, 3 of 16 samples (i.e., ZLR-08, ZLR-11, and ZLR-12) detected three or more clonal integration events. These distinct integration events in the same sample tended to be clustered together and partially overlapped with different breakpoints. The Nanopore long-read sequencing technology used in this study is amplification-free and allowed direct, long-read sequencing of native DNA, thereby eliminating any potential PCR bias. Each chimeric read likely represents a single cell within the sequenced bulk cells. It is less likely that these integration events simultaneously occurred in a diploid cell since there are generally only two copies of a specific chromosomal site for viral integration. Second, some of these events shared the use of the same integration breakpoints within the same tumor. It is also less likely that these distinct integration events occurred in different cells independently yet produced the same breakpoint. A reasonable inference is that these integration events are evolutionarily related, and some integration events may derive from a primary integration event in a subset of its clonal cells. Understanding clonal evolution related to HPV integration is of great significance in revealing the mechanism of cervical tumor progression and metastasis which could then lead to the development of new cancer therapeutic strategies.

Characterization of integrated HPV DNA segments in the host genome will enable a better understanding of the pathogenic mechanism and the role of HPV in cervical cancer. Several forms of viral DNA integrated into the host genome have been previously proposed, including a single viral genome, multiple head-to-tail tandemly repeated viral genomes, and tandem copies of the HPV genome interspersed with host DNA[19,51,52]. However, current knowledge of viral integration in cervical tumors was mostly based on Southern blot and short-read sequencing analyses. These analyses yielded limited characterization of the complete virus-human integration events in cervical cancer. Using long-read sequencing technologies, we accurately characterized four distinct types of integrated HPV DNA segments that are frequently present in clonal integration events: Type A, a truncated HPV genome containing *E6/E7*; Type B, a truncated HPV genome lacking *E6/E7*; Type C, an overflowing continuous segment containing the intact HPV genome; and Type D, a combination of Type A, Type B or Type C. Of note, Type B, which lacks *E6/E7* genes previously considered to be the main causes for cervical cancer, has been not reported in the literature to our knowledge. Despite this, Type B was not very rare in our dataset where 5 out of 22 detected integration events contained the integration Type B. On the other hand, based on our dataset (Fig. 3), the Type B integration events were not exclusively presented in tumors, indicating the Type B integration event may

be a "passenger event" during cervical disease progression. However, we also observed that some Type B integration events contain oncogenes of the host genome (such as *CCAT2* in ZLR-11), thereby likely promoting cervical carcinogenesis independently of *E6/E7* oncogenesis. These remain to be further studied. In addition, our long-read sequencing analysis provides a high-resolution structure of these complex HPV integrations. This is best exemplified by the sample ZLR-02 where two different disrupted HPV16 genomes are interspersed with 5-bp cellular sequences and integrated at a single locus. Such an occurrence could not have been detected by previous methods such as Southern blot and short-read sequencing analyses.

Dysregulated expression of *E6/E7* genes upon HPV integration is frequently observed in HPV-associated cancers. It is this that has been thought to be a primary trigger of the oncogenic progression by disturbing the cell cycle and inducing genomic instability in cervical cancer[14]. Secondary genetic events in the host genome, resulting from HPV integration, are also essential for the development and progression of cervical cancer. Combined with the corresponding RNA-seq data, we found 9 genes with significantly altered expressions near integration breakpoints. We further highlighted *LINC00290, LINC02500,* and *LENG9* as potential driver genes in cervical cancer. Especially in the case of *LENG9*, we revealed the virus-human hybrid ECC structure, which can replicate autonomously and lead to subsequent amplification, possibly as a potential mechanism for the upregulation of target genes. We also performed functional analysis of *LEGN9* using gain- and loss-of-function assays and validated its oncogenic role in cervical cancer. *LENG9* is rarely reported in the literature. GO function analysis revealed that the molecular function of LENG9 is metal ion binding (GO:0046872), and the BioGRID database[53] showed that LENG9 has 6 interactors, including C9ORF41, B4GALT2, CDC5L, D2HGDH, FOXS1 and OGT (https://thebiogrid.org/125106). In particular, B4GALT2 is also annotated to GO term of metal ion binding and its expression is dramatically reduced in tumor ZLR-08 compared to adjacent normal tissue. It has been reported that depletion of B4GALT2 inhibits p53-mediated cell apoptosis in Hela cells[54]. Furthermore, CDC5L (alias HCDC5) has been reported as a positive regulator of cell cycle G2/M progression[55]. Their interactions with LENG9 may be a potential mechanism underlying the promotion of cell proliferation and migration when LENG9 is upregulated in cervical cancer cells while these warrant further functional investigation in the future.

Several caveats of our study should be acknowledged. Firstly, in our study, only a proportion of the breakpoints were involved in the reported integration events. In this, it may be considered that other rare events had probably occurred but were not found, despite our sequencing depth reaching >30×. The inability to exhaustively detect all of these events may be partly due to the sequencing read length still not being sufficiently long. In sample ZLR-11, for example, whilst in addition to the four clonal

integration events validated, we also identified seven additional events with one supporting chimeric read (Supplementary Fig. 10). However, due to the inability to design PCR primers or the weak bands of PCR products, one of the breakpoints in these rare events remained unverified. Further investigations such as larger depth and longer reads of sequencing may be required for validating these rare clonal integration events. Second, to efficiently identify genes that may be affected by HPV integration, we focused on those genes that are within or closest to HPV breakpoints. The trade-off is that some true target genes outside this range may be excluded. Similarly, to explore candidate oncogenes, we focused on those genes whose expression was significantly altered. For example, in addition to *LENG9*, two other genes, *LENG8* and *CDC42EP5*, were also affected by HPV-related aberrant genomic events in ZLR-08 (Fig. 6a), although no significant expression changes were observed. However, their potential effects cannot be completely eliminated, and they may be secondary candidates for the oncogenic event in ZLR-08. Third, it has been reported that although their carcinogenic risk is slightly different, a variety of high-risk HPV types (such as HPV16, HPV18, and HPV58) can lead to the development of cervical cancer. The present study only focused on HPV16-positive cervical tumors. Future research needs to be extended to investigate large-range virus-human integration events in other high-risk HPV types such as HPV18 and HPV58.

In summary, here we have depicted large-range virus-human integration events at high resolution and revealed potential intratumor heterogeneity of HPV integration in cervical tumors. We further identified four distinct types of integrated HPV DNA segments that are frequently present in clonal integration events. Multiple clonal integration events involved the use of shared breakpoints, induced inter-chromosomal translocations, and formed ECC DNA structures. Combined with the corresponding RNA-seq data, we highlighted *LINC00290*, *LINC02500,* and *LENG9* as potential driver genes in cervical cancer. Finally, we revealed the spatial relationship between HPV integration and various SVs as well as their functional consequences in cervical cancer. All of these findings provide insight into the HPV integration and its oncogenic progression in cervical cancer.

## Methods

**Sample acquisition and processing**. The study was approved by the Institutional Review Board of the Women's Hospital of Zhejiang University (Hangzhou, China). The study was conducted in accordance with the International Ethical Guidelines for Biomedical Research Involving Human Subjects. All subjects provided informed consent to participate in the study. All tumor tissues were obtained at the time of diagnosis before any treatment was administered. Fresh tissues were collected, snap-frozen in liquid nitrogen, and stored at −80 °C. Clinical information (e.g., tumor type and stage) was extracted from pathology reports. A review of hematoxylin and eosin (H&E) slides was performed by a gynecologic pathologist to confirm the diagnosis. Tumor cell content of sections from frozen cervical tissue samples embedded in optimal cutting temperature medium was confirmed to be > 70% by H&E staining.

**HPV typing**. Genomic DNA was extracted from 25 mg of the tissue using Pure-Link® Genomic DNA Kit (Life Technology, Inc). The DNA concentration of each sample was measured using Nanodrop 2000 (Thermo Fisher Scientific, Wilmington, DE) and DNA integrity was assessed using agarose gel electrophoresis. DNA samples were subjected to PCR using specific primers of HPV16 E6. PCR was performed using a ProFlex™ PCR thermal cycler. The 16 HPV16 E6 positive samples were then prepared for Nanopore sequencing.

**Library construction and Nanopore sequencing**. Sequencing libraries were constructed following the instructions from Oxford Nanopore Technologies (ONT) for 1D ligation sequencing on PromethION. Firstly, the quality of input genomic DNA was checked and 2 μg high molecular weight genomic DNA was sheared into long fragments with an average size of 8 kb using g-TUBE (Covaris). These DNA fragments were then repaired, end-repaired, A-tailed, and adapter-ligated. DNA repair was conducted using NEBNext FFPE DNA Repair Mix (M6630L, NEB). DNA end repair and dA-tailing were performed using NEBNext

Ultra II End Repair/dA-Tailing Module (E7546L, NEB). Adapter ligation was performed using the NEBNext Blunt/TA Ligase Master Mix (M0367L NEB) and a Ligation Sequencing Kit 1D (SQK-LSK109). DNA was purified between each step using Agencourt AMPure XP beads (A63882, Beckman Coulter) and was quantified via a Qubit Fluorometer 2.0 (ThermoFisher Scientific, Waltham, MA). The prepared libraries were sequenced using a PromethION sequencer and 1D flow cell with protein pore R9.4.1 1D chemistry following the manufacturer's guidelines. The Nanopore sequencing results were basecalled using the ONT's basecaller Guppy (v4.0.11), converting the current intensity values (in fast5 format) to nucleic acid sequences (in fastq format). The original fastq data were further filtered to remove adapters, short reads (length < 500 bp), and low-quality reads (mean base quality < 6), and then the total cleaned sequence data was obtained (Supplementary Table 1).

**Long-read sequence alignment**. The clean long reads were aligned to a custom in-house reference genome that comprises the human genome (GRCh37/hg19; https://hgdownload.soe.ucsc.edu/goldenPath/hg19/chromosomes/) and multiple HPV genomes using the long-read mapper NGMLR (v0.2.7; https://github.com/philres/ngmlr)[36]. The output sequence alignment results in SAM format were further transformed to BAM format and sorted by genomic coordinates using samtools (v1.9; http://samtools.sourceforge.net/)[56] for subsequent SV identification and breakpoints detection.

**SV and breakpoint detection**. The long-read SV caller Sniffles (v 1.0.11; https://github.com/fritzsedlazeck/Sniffles)[36] was employed to detect all types of SVs (such as deletions, duplications, inversions, and translocations) with three minimum supporting reads. The SV type breakends (BND) involving HPV were extracted and the junction position of human and HPV sequence was reported as the breakpoint for HPV integration. The breakpoint positions on the human genome were then extracted and annotated using ANNOVAR[57]; while corresponding positions on the HPV16 genome were annotated using an in-house program based on the latest HPV16 reference sequence with gene annotation (7906 bp) from the Papilloma Virus Episteme (PaVE) (https://pave.niaid.nih.gov).

**Detection of HPV clonal integration events**. Although the alignment results from NGMLR already contained partial integration event information, these were not sufficient to identify HPV clonal integration events. Therefore, we re-analyzed chimeric reads that span both human and virus genomes using BLAST+ (v2.10.1; ftp://ftp.ncbi.nlm.nih.gov/blast/executables/blast+/2.10.1/). First, chimeric reads were extracted from the NGMLR alignment results using samtools[56]. These read sequences were then realigned to the custom in-house reference genome as described above, using megablast of local BLAST+ (*E* value < 1.0e-20). Finally, using in-house Perl scripts, these chimeric reads containing at least one intact integration event (a segmental HPV DNA surrounded by a human genome on both sides) were further analyzed on a case-by-case basis. Some representative clonal integration events were visualized using the R statistical package (v3.6.2; https://www.r-project.org/).

Somatic SVs were identified by comparing the SV results from tumor samples to those from six nonmalignant cervix tissues. First, detected SVs from normal samples were pooled using SURVIVOR (v 1.0.7)[58]. Then, somatic SVs were identified for each tumor sample by SURVIVOR in comparison to the pooled SVs from normal samples.

**SV annotation and analysis**. SV types including BND, deletion (DEL), duplication (DUP), insertion (INS), inversion (INV), and inverted duplication (INVDUP) were detected using Sniffles[36]. Prior to SV annotation and analysis, we filtered SVs using the following criteria: (1) removal of variants covered by less than three reads to reduce the false negative, (2) removal of SVs whose length < 50 bp, (3) removal of non-canonical SV types including DEL/INV, DUP/INS and INV/INVDUP. AnnotSV (v3.0.2; https://github.com/lgmgeo/AnnotSV)[59] was then used to annotate filtered SV and provide gene-level information.

We divided the reference genome into non-overlapping windows of 1 kb, summarized each genomic feature per window, and then mapped the summarized metrics to each variant using bedtools (v2.24.0; https://github.com/arq5x/bedtools2)[60]. The replication time metric was calculated as averaged Wavelet-smoothed signal value per window by Repli-Seq data in HeLa-S3 cell line from ENCODE[61]. A TAD boundary was obtained from the study of Dixon et al.[62], where the distance to nearest boundary was normalized as log2(kb of distance + 1). A recombination value was obtained from the HapMap Project[63], with the value at the nearest point taken for annotation. The core 15 chromatin states were then downloaded from the Roadmap Epigenomics Project[64].

**RNA-seq analysis**. RNA-seq libraries were prepared for the same set of cervical tumor tissues using the NEBNext Ultra RNA Directional Library Prep Kit according to the manufacturer's instructions. Libraries were multiplexed and sequenced on a HiSeq X10 sequencer with 150 bp paired-end reads (Illumina, San Diego, CA). Briefly, adapters and low-quality (base quality < 13) sequences were trimmed from the output reads (Supplementary Table 5). The trimmed short sequence reads were aligned to the custom in-house reference genome as described

above using HISAT2 (v2.2.1) with mammalian default parameters[65]. Transcript construction, quantification, and normalization of transcript abundance were performed using StringTie2 (v2.1.5)[66].

**Expression analysis**. To evaluate whether there was upregulation of genes upon HPV integration, we estimated the $z$ score of each disrupted gene in the integrated sample among all tumor samples. Then the one-sided significant level was calculated from a standard normal distribution and adjusted by the FDR. The RNA-seq data of 16 cervical tumor tissues from the present study together with 87 tumor tissues and 39 adjacent normal tissues from our recent study[41] were used for expression analysis.

**PCR validation of virus-human breakpoints**. To validate the breakpoints detected by long-read sequencing, PCR and Sanger sequencing were further performed. Based on the mapped sequencing reads, PCR primers were designed on the left and right flanking sequences of the target junction site (Supplementary Table 4). The PCR products were assessed using agarose gel electrophoresis. PCR was performed using a ProFlex™ PCR thermal cycler (Life technology, Carlsbad, CA). PCR products were then purified and sequenced using the Sanger sequencing method. The primers for PCR reactions were used for sequencing. Due to low-frequency clonal integration events, some faint bands at the right size of PCR product were further analyzed using nested PCR.

**PCR validation of ECC DNA structure**. In brief, a 10 μl control plasmids mixture was applied for 300 ng genomic DNA extracted from each sample. The plasmid stock mixture consisted of three plasmids in different concentrations: pBR322 at 48 ng, pUC19 at 1.5 ng, and pUG72 at 0.02 ng. The linear DNA was removed by exonuclease (Plasmid-Safe ATP-dependent DNase, Epicentre, Madison, WI, USA) at 37°C in a heating block and the digestion was carried out continuously for 5 days, adding additional ATP and DNase every 24 h according to the manufacturer's protocol. Then, the exonuclease was heat-inactivated at 70°C for 30 min. The complete removal of linear DNA was confirmed by qPCR of the COX5B gene.

PCR oligos were designed using Primer3web (version 4.0) to yield products across junctions of ECC DNA evaluated in the study (Supplementary Table 7). Each 50 μl PCR reaction typically included 100 ng rolling-circle amplification template, 10 μM primers, dNTP, 25 mM MgSO4, buffer, and KOD Hot Start DNA polymerase (Novagen), and PCR was for 35 cycles in a PCR cycler under standard PCR conditions. All reactions were performed with controls of human genomic DNA. Three internal plasmids (pBR322, pUC19, and pUG72) were positive controls for PCR reaction with circular DNA templates. The PCR products were separated on 1.0% agarose gel electrophoresis.

**Cell culture**. Human HPV-positive cervical cancer cell lines (CaSki and SiHa) were purchased from the American Type Culture Collection (ATCC) (Manassas, VA, USA). CaSki cells were cultured in RPMI-1640 medium (Gibco, Carlsbad, CA, US). SiHa cells were cultured in MEM medium (Gibco). All medium was supplemented with 10% fetal bovine serum (FBS, Gibco) and 1% penicillin/streptomycin solution (Gibco). The two cell lines were maintained in a humidified atmosphere with 5% CO$_2$ and at 37°C.

**Cell transfection**. For overexpression assay, the full-length human LENG9 sequence (1506 bp, NM_198988.1) was obtained by PCR and subcloned into the PLVX-puro vector by Lipofectamine 3000 (Invitrogen, Carlsbad, CA, USA) to establish cells stably overexpressing LENG9, whereas the mock vector without LENG9 sequence was used as a control. For knockdown assays, three short interfering RNAs (siRNAs) targeting LENG9 (Supplementary Table 7) were synthesized by GenePharma (Shanghai, China). Cells were seeded at ~50% confluency and transfected using GenMute™ siRNA transfection reagent (SignaGen™ Laboratories, Rockville, MD, USA) at a final concentration of 10 μM following the manufacturer's instructions. The transfection efficiency of LENG9 in cells was determined by western blot with antibody LENG9 (Proteintech #16295-1-AP, 1:500). GAPDH antibody was used as a loading control antibody (Proteintech #60004-1-Ig, 1:20000).

**RNA isolation and quantitative RT-PCR**. RNA was isolated from cells using Trizol reagent (Invitrogen, Carlsbad, CA, USA) following the manufacturer's instructions. Reverse transcription was performed using HiScript® II Q RT SuperMix (Vazyme, Nanjing, China). Quantitative real-time (RT) PCR was performed on the cDNA using the ChamQ Universal SYBR qPCR Master Mix (Vazyme). GAPDH was used as an endogenous control for normalizing the expression of tested genes.

**Cell proliferation, migration, and invasion assays**. Cell proliferation assays were performed using the Cell Counting Kit 8 (CCK8; MCE, Monmouth Junction, NJ, USA). First, cells were seeded at a density of 1000 cells/well in 96-well plates containing 100 μL of the medium. Then, OD450 was measured 2 h after adding 10 μL of CCK8 solution. Cell migration and invasion assays were performed using Transwell chambers (Corning Costar, Tewksbury, MA, USA) following the

manufacturer's instructions. Cells were transfected and cultured with a serum-free medium for 24 h. Next, these cells were detached and resuspended in a serum-free medium. Then, these suspended cells were added to the upper transwell chambers for migration and invasion assays. At the same time, RPMI-1640 medium (for CaSki) or MEM medium (for SiHa) supplemented with 10% FBS was added to the bottom chamber. After an appropriate time to migrate or invade, the cells that migrated into the bottom chamber were fixed with 100% methanol and stained with 0.1% crystal violet staining solution. Images were captured from each membrane and the number of migratory cells was counted under a microscope (Leica DM4000, Buffalo Grove, IL, USA).

**Statistical analysis**. Data are presented as mean ± standard deviation of three or more independent experiments. A two-sided Student's $t$ test was used to analyze the differences between groups. KEGG pathways and GO terms that were enriched for genes potentially disrupted by HPV integration were detected using Metascape (https://metascape.org/gp/index.html/main/step1). Fisher's exact test was used for assessing the association of integration features with clinical consequences. All statistical analyses were implemented in the statistical package R (v3.6.2; https://www.r-project.org/).

**Reporting summary**. Further information on research design is available in the Nature Research Reporting Summary linked to this article.

## Data availability

The raw RNA-seq data (fastq files) generated in this study have been deposited in the Genome Sequence Archive (GSA) under the accession number HRA001925. To protect patients' genetic and genomic privacy, all data requests shall be made through GSA, to be evaluated and approved by the appointed Data Access Committee (DAC). For detailed guidance on making the data access request, see GSA-Human_Request_Guide_for_Users [https://ngdc.cncb.ac.cn/gsa-human/document/GSA-Human_Request_Guide_for_Users_us.pdf]. The approximate response time for accession requests is 30 working days. The reference human genome (GRCh37/hg19) was downloaded from the UCSC genome Browser (https://hgdownload.soe.ucsc.edu/goldenPath/hg19/chromosomes/). The HPV16 reference sequence with gene annotation (7906 bp) was downloaded from the Papilloma Virus Episteme (https://pave.niaid.nih.gov). All other data supporting the findings of this study are available within the Article, Supplementary Data, and a data source file. A data source file is provided with this paper. Source data are provided with this paper.

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

## Acknowledgements

We thank the Core Facility at Zhejiang University School of Medicine for providing technical support, Christopher R. Wood for reading and commenting on the manuscript. This work has been supported in part by the National Key R&D Program of China (2021YFC2701204, H.W.), the Key Program of Zhejiang Provincial Natural Science Foundation of China (LZ20H160001, Y.L.), Key R&D Program of Zhejiang Province of China (2021C03126, Y.L.), Medical Health Science and Technology Key Project of Zhejiang Provincial Health Commission (WKJ-ZJ-2007, Y.L.), National Natural Science Foundation of China (82072857 and 82188102, Y.L., P.L.), and the Leading Innovative and Entrepreneur Team Introduction Program of Zhejiang (2019R01001, Y.L.).

## Author contributions

Y.L., P.L., and W.L. considered and designed the study. L.Z. and Q.Q. developed the algorithm and performed the data analysis. Q.Z., J.L. M.Y., and L.X. conducted validation experiments and in vitro cell-based assays. H.X. and M.Y. assisted in data analysis and quality control. W.L., K.L., H.W., and X.K. collected tumor tissues. L.B. performed a

pathological analysis of tumor samples. L.Z., Q.Q., Y.L., and P.L. wrote the manuscript. All of the authors commented and approved the study.

## Competing interests

The authors declare no competing interests.
