## [Peer review file · Nature Communications]

REVIEWER COMMENTS

Reviewer #1 (Remarks to the Author): Expert in long-read sequencing, cancer genomics and bioinformatics

In this manuscript, Liyuan Zhou et al. analyzed HPV integration in cervical cancer using nanopore sequencers. They performed long read sequencing of 16 HPV16-positive cervical cancers and identified and characterized host-viral integration breakpoints and clonal integration events by taking advantages of long read sequencing technologies. They defined four types of clonal integration events; Type A - D. They also analyzed gene disruption caused by HPV integration and identified potential driver genes having oncogenic roles, such as CDC42EP5. They showed that HPV integration-related SVs have great impact to cervical carcinogenesis. I think that clear association between each feature of integration events and clinical consequences of cervical cancers is needed to be additionally characterized. There are also some points that need to be addressed to improve the manuscript as below;

Points;

1. The authors associated HPV integration with gene disruption (line 227, p. 8). Please describe the reason why genes around 500 kb near the integration breakpoints were extracted. I wonder if integration events can affect genes 500 kb apart.
2. The authors characterized the overall SV landscape of the host genome in cervical cancer (line 277, p. 9). Please describe whether the detected SVs were somatic (not polymorphisms?). I think the authors need to focus somatic SV events when they analyze association between HPV integration and SV occurrence.
3. In Kaplan-Meier analysis using the TCGA data, the authors showed association between CDC42EP5 expression and prognosis. The authors should explain what extent contribution of HPV integration events to alteration of the gene expression in cervical cancer population. They also need to describe potential mechanism that CDC42EP5 expression was upregulated by HPV integration.
4. The authors mentioned "substantial intratumor heterogeneity" in the abstract section. However, I could not find sufficient explanations of clone patterns and impact of heterogeneity to oncogenic progression and patient prognosis in cervical cancer. I think that, for example, variant allele frequency of integration breakpoints and surrounding mutations in the host genome would be informative for understanding clone architecture.

Minor points;

1. In Fig. 2A, arrows are out of position.

Reviewer #2 (Remarks to the Author): Expert in HPV and cervical cancer genomics

The Nanopore long-read sequencing technology has opened new avenues in research related to HPV-DNA integration. Zhou and colleagues present a well written paper highlighting novel findings which are clearly of interest for experts in this field. Largely the conclusions drawn are supported by the data provided but several aspects require clarification:

Page 2, lines 54 to 56 state: "Our study depicts large-range virus-human integration events at high resolution with several novel characteristics and reveals substantial intratumor heterogeneity of HPV integration in cervical carcinogenesis."

In this sentence the term intratumor heterogeneity is misleading. The authors need to define in their manuscript what is actually meant by intratumor heterogeneity. As reported in several previous papers most tumors harbor more than one HPV integration event. If this is meant by intratumor heterogeneity their findings would be of confirmative nature. It would, however, be of interest to know whether the integration events have occurred within the same tumor cell or are scattered throughout the tumor giving rise to different subclonal tumor cell populations. The authors provide no evidence for this type of intratumor heterogeneity. Strong evidence for the lack of intratumor heterogeneity in terms of distribution of integration events throughout the tumor tissue is provided by Carow et al. 2017 (doi:10.3390/ijms18102032). These findings need to be discussed.

Page 3, lines 77 to 79 state: "Upon integration, overexpression of viral oncogene E6 and/or E7 is frequently observed, which may result from the disruption of their negative regulators E1/E2, and the increased stability of E6 and/or E7 mRNAs expressed from the viral-host fusion transcripts."

Häfner et al. 2008 (doi: 10.1038/sj.onc.1210791) provide evidence that HPV16 DNA integration does not invariably result in high levels of oncogene transcripts. Moreover, the transcript levels between CIN (mainly high-grade lesions) and cervical carcinomas were shown to be similar. Concerning transcript stability the work by Ehrig et al. 2020 (doi:10.3390/ijms21010112) provides further insights and should cited.

Page 3, lines 90 to 91 state: “Using NGS technologies, several studies have been carried out to characterize genome-wide alterations in both viral and host genomes upon HPV integration in cervical cancer. “

A seminal contribution for the detection of HPV integration sites using NGS technology was also made by Xu et al. 2013 (doi: 10.1371/journal.pone.0066693) and should be cited.

Page 8, lines 230 to 232 state: “Genes at HPV hot spot integration sites included KLF5, LINC00392, CASC8, CASC21, LINC00290 and MACROD2. In particular, the KLF5-LINC00392 on chromosome 13 was identified at a recurrent integrated site with 5 breakpoints in two tumors.”

The term hot spot is misleading. In previous publications the term “hot spot” refers to common chromosomal regions which are affected by HPV integration in several independent tumors (see Schmitz et al 2012, doi: 10.1371/journal.pone.0039632 and Hu et al 2015, doi:10.1038/ng.3178). This then poses the question as to which integration events of the current study map to previously reported hot spots?

Page 9, lines 253 to 254 state: “We particularly focused upon two HPV-integrated tumors (ZLR-01 and ZLR-08) with low expression of E6 and E7 whose upregulation had been considered to be primarily responsible for cervical carcinogenesis.”

Figure 5A and 5B show the expression level of E6 and E7 of 103 cervical carcinomas, respectively. Expression in sample ZLR-08 is highlighted by a red dot. It is rather surprising that a large proportion of cervical cancers examined have such low levels of E6/E7 RNA. There seems to be a 6 fold log₂ difference in transcription levels. This is not in line with the general understanding that cervical cancers have high E6 and E7 transcript levels. This should be explained.

Page 12, lines 379 to 382: “Of note, Type B is a novel type of integrated HPV DNA segment in the host genome, which lacks E6/E7 genes that have been previously considered to be the main causes for cervical cancer. Despite this, Type B was common, observed in 2 of the 12 tumors detected with clonal integration events.”

Can “2 of 12 tumors” be considered to be a common event? Moreover, Type B integrated fragments do not seem to be the only integration event in the respective tumors. These tumors seem to also harbor E6/E7 containing integration events (see Figure 3).

Page 13; lines 387 to 389 state: “Overexpression of E6/E7 genes upon HPV integration is frequently observed in HPV-associated cancers. It is this that has been thought to be primary trigger of the oncogenic progression by disturbing the cell cycle and inducing genomic instability in cervical cancer.”

As already mentioned above, DNA integration does not invariably result in high levels of oncogene transcripts. Moreover, the transcript levels between CIN (mainly high-grade lesions) and cervical

carcinomas were shown to be similar. Concerning transcript stability the work by Ehrig et al. 2020 (doi:10.3390/ijms21010112) provides further insights and should be cited.

Below are responses to comments from the two Reviewers. We greatly appreciate their constructive and thoughtful comments and suggestions, which significantly improved and clarified the presentation of our paper.

Please note that *the comments are shown in italics and bold font*. Figures R1-R2 and Tables R1-R2 are new displayed items for reviewers only that were generated from our recent data analyses in response to reviewers' comments. The other figures referred to in the responses can be found in either the main text or supplementary data of our manuscript.

Major changes made in the revised manuscript were highlighted in **red font**.

Reviewer #1 (Remarks to the Author): Expert in long-read sequencing, cancer genomics and bioinformatics

General Comments:

In this manuscript, Liyuan Zhou et al. analyzed HPV integration in cervical cancer using nanopore sequencers. They performed long read sequencing of 16 HPV16-positive cervical cancers and identified and characterized host-viral integration breakpoints and clonal integration events by taking advantages of long read sequencing technologies. They defined four types of clonal integration events; Type A - D. They also analyzed gene disruption caused by HPV integration and identified potential driver genes having oncogenic roles, such as CDC42EP5. They showed that HPV integration-related SVs have great impact to cervical carcinogenesis. I think that clear association between each feature of integration events and clinical consequences of cervical cancers is needed to be additionally characterized. There are also some points that need to be addressed to improve the manuscript as below;

Response:

Thank you for your valuable comments. We have collected the follow-up information of 16 patients (revised **Table 1**) and explored the association between integration features and clinical consequences. To date, two patients have died of local relapse or distant metastases of cervical cancer. One patient (ZLR-08) was completely out of touch with us and her status was unavailable. Thus, based on a dataset of 15 patients, a significant association between multiple integration events (≥ 2) and poor prognosis (decease) was observed (Fisher's Exact Test; $P = 0.029$). Furthermore, as discussed in our response to comment #4 below, it was these two deceased patients who exhibited some intratumor heterogeneity, which may account for their poor prognosis (Fisher's Exact Test; $P = 0.009$). While we did not find evident associations between other integration features (such as integration types) and clinical consequences in this relatively small dataset (pages 6-7).

Table 1. Characteristics of the 16 tumor samples and the number of breakpoints and integration events detected in these samples.

Patient-ID	Age at diagnosis	HPV type	Stage	Breakpoints	Events	E6	E7	Follow-up (Months)	Status at the end of follow-up
ZLR-01	45	16	stage I	2	1	2.9	8.1	49	tumor-free
ZLR-02	51	16	stage III	2	1	140.8	460.2	30	tumor-free
ZLR-03	57	16	stage II	3	1	67.5	177.5	28	tumor-free
ZLR-04	50	16	stage I	3	1	91.4	212.8	63	tumor-free
ZLR-05	47	16	stage II	0	0	52.3	146.4	51	tumor-free
ZLR-06	34	16	Stage II	13	2	320.5	683.4	88	tumor-free
ZLR-07	45	16	Stage I	2	0	115.1	236.4	90	tumor-free
ZLR-08	44	16	Stage I	12	5	504.3	1213.7	NA	NA
ZLR-09	37	16	Stage II	2	0	703.9	1261.5	61	tumor-free
ZLR-10	50	16	Stage I	1	0	12.9	20.3	20 + 44	relapse + tumor-free
ZLR-11	50	16	Stage II	10	4	600.1	1251.1	14 + 3	relapse + deceased
ZLR-12	47	16	Stage II	4	3	271.6	539.2	54 + 18	relapse + deceased
ZLR-13	52	16	Stage II	3	1	777.2	1719.0	90	tumor-free
ZLR-14	45	16	Stage II	2	1	162.9	285.6	60	tumor-free
ZLR-15	50	16	Stage II	2	1	152.1	218.0	62	tumor-free
ZLR-16	58	16/58	Stage II	2	1	73.0	235.6	92	tumor-free

Note: Histologic subtype of all 16 samples is squamous cell carcinoma (SCC); NA indicates data are not available. Expression levels of E6/E7 are quantified by transcript per million (TPM) in RNA-seq data.

Points:

1. The authors associated HPV integration with gene disruption (line 227, p. 8). Please describe the reason why genes around 500 kb near the integration breakpoints were extracted. I wonder if integration events can affect genes 500 kb apart.

Response:

Thank you for your rigorous consideration. HPV integration can disrupt cellular genes or their flanking sequences, altering their expression or the expression of nearby genes (*PLoS Pathog.* 2017, 13(4):e1006211). To characterize genes potentially disrupted by HPV integration, we annotated genes around 500 kb near the integration breakpoints using ANNOVAR (*Nucleic Acids Res.* 2010, 38(16):e164). The distance threshold of 500 kb for annotating genes around integration sites has been used in several previous studies (e.g., *Nat Genet.* 2015, 47(2):158-63; *Genomics.* 2021, 113(3):1554-1564; *Genome Res* 2022, 32(1): 55–70.).

However, we completely agree with you that integration can potentially affect genes 500 kb apart. HPV integration may also affect cis-acting regulators of genes, which can influence their target genes over longer distances. The distance threshold of 500 kb was an empirical but somewhat arbitrary value in previous studies. Using a longer distance of threshold will increase the chance of including these potentially affected genes, but also increase the risk of adding more unrelated genes. Considering the trade-off between false positives and false negatives, a threshold of 500 kb was used in this study.

2. The authors characterized the overall SV landscape of the host genome in

cervical cancer (line 277, p. 9). Please describe whether the detected SVs were somatic (not polymorphisms?). I think the authors need to focus somatic SV events when they analyze association between HPV integration and SV occurrence.

Response:

Thank you for your thoughtful suggestion. Our previous SV landscape characterization was based on all SVs detected from tumors. To control the genomic background, we mainly focused on genome windows spanning HPV integration sites. We then performed a comparative analysis of SVs from regions with or without HPV integration. Upon your advice, we performed additional long-read sequencing on adjacent normal tissues (with available high-quality of DNA) from 6 of 16 tumors. Following our standard analysis pipeline, SVs detected from normal samples were pooled together using SURVIVOR, which can thus benefit all tumors. Somatic SVs were identified for each tumor sample by SURVIVOR against the pooled SVs from normal samples. The somatic SV landscape was then re-characterized. The relationship between HPV integration and elevated SV occurrence remained unchanged. In terms of genomic features and functional implications of HPV integration-related somatic SVs, an overall similar result with minor difference was reported. The detailed description has been updated in the revised manuscript (pages 11-12).

3. In Kaplan-Meier analysis using the TCGA data, the authors showed association between CDC42EP5 expression and prognosis. The authors should explain what extent contribution of HPV integration events to alteration of the gene expression in cervical cancer population. They also need to describe potential mechanism that CDC42EP5 expression was upregulated by HPV integration.

Response:

We gratefully appreciate for your valuable suggestion. When we attempted to explore the mechanism of upregulation of CDC42EP5 in sample ZLR-08 using RNA-seq data, we unfortunately recognized that the tumor and normal tissues of ZLR-08 were swapped by mistake. Initially, we observed that the expression level of HPV16 E6/E7 was much higher in ZLR-08_N (labeled normal) than that in ZLR-08_C (labeled cancer). Notably, the number of detected HPV-human junction sites in the RNA-seq data of ZLR-08_N was also greater than that in the RNA-seq data of ZLR-08_C. Therefore, we suspected that ZLR-08 tumor and matched adjacent normal tissues were erroneously swapped, which was subsequently confirmed by tissue H&E staining. Four breakpoints were detected in the DNA sample of ZLR-08_C (now relabeled as adjacent normal tissue) as it contained a small proportion of tumor tissues. This was also supported by the fact that 2 out of 4 breakpoints could be only verified by nested PCR in ZLR-08_C, indicating the low concentration of HPV-related fragments in ZLR-08_C. Finally, the labeled “normal” DNA sample (i.e., ZLR-08_N) was sent to long-read sequencing for validation and analysis. It turned out that there were 12 breakpoints and 5 integration events detected in this labeled normal

sample (compared with 4 breakpoints and 2 integration events detected in ZLR-08_C), further confirming ZLR-08_N as a tumor tissue. We sincerely apologize for the mistake.

We then performed the analysis based on the correct tumor sample. In the revised ZLR-08, there were 12 breakpoints detected, of which 6 were involved in the 5 detected integration events, with 1 breakpoint being commonly used in all 5 integration events (**Table R1**). In addition, all 12 breakpoints were clustered in 19q13.42. Accordingly, the genomic region associated with these 5 integration events contained three genes: LENG8, LENG9 and CDC42EP5 (**Table R2**). Based on the DNA sequencing base coverage around this target region (new **Fig. 6A**), two significant focal amplifications were observed, corresponding to human fragments in ZLR-08_e1/e4 and ZLR-08_e2, respectively (**Table R2**). Interestingly, all these three integration events could potentially form virus–human hybrid ECC structure, which can replicate autonomously and lead to amplification. Whereas the other two events ZLR-08_e3 and ZLR-08_e5 occurred less frequently according to the sequencing depth of their involving breakpoints (**Table R1**). This could be the explanation for the absent manifestation of these two events on the DNA depth plot (new **Fig. 6A**). Furthermore, it was observed that only the gene LENG9 was completely amplified in this target region and the corresponding expression level is highest among all 103 cervical tumors that we sequenced in this and our previous studies. Its expression was increased up to an approximate 10-fold compared to the average expression level of other 102 tumors (new Fig.6B; z-score=17.4; P-value=1.98e-68).

Table R1. 12 detected breakpoints and their involving events in revised ZLR-08

Patient-ID	Breakpoints	Events involved	GT:DR:DV
ZLR-08	Chr19:54983022_HPV16:2274	@	1/1:1:22
ZLR-08	Chr19:54974840_HPV16:2636	e2	1/1:0:45
ZLR-08	Chr19:54970568_HPV16:7029	e1	0/1:18:19
ZLR-08	Chr19:54974755_HPV16:7906	@	0/0:19:3
ZLR-08	Chr19:54984924_HPV16:2024	@	0/1:4:15
ZLR-08	Chr19:54978308_HPV16:3050	e3	1/1:0:5
ZLR-08	Chr19:54970599_HPV16:1686	e4	1/1:0:18
ZLR-08	Chr19:54983028_HPV16:2276	e1/e2/e3/e4/e5	1/1:0:84
ZLR-08	Chr19:54974755_HPV16:131	@	0/0:19:3
ZLR-08	Chr19:54974756_HPV16:6486	@	1/1:0:4
ZLR-08	Chr19:54979846_HPV16:4968	@	1/1:0:4
ZLR-08	Chr19:54978038_HPV16:1642	e5	1/1:0:4

GT: genotype; DR: number of high-quality reference reads; DV: number of high-quality variant reads

Table R2. Five detected integration events and their associated genes in revised ZLR-08

Chr	Start	End	Distance	Type	Region	Genes	Events
19	54970568	54983028	12460	DEL/ECC	exonic	CDC42EP5;LENG8;LENG9	ZLR-08_e1
19	54974840	54983028	8188	DEL/ECC	exonic	CDC42EP5	ZLR-08_e2
19	54978308	54983028	4720	UD	UTR5	CDC42EP5	ZLR-08_e3
19	54970599	54983028	12429	DEL/ECC	exonic	CDC42EP5;LENG8;LENG9	ZLR-08_e4
19	54978038	54983028	4990	DEL/ECC	UTR5	CDC42EP5	ZLR-08_e5

DEL, deletion; ECC, extrachromosomal circular DNA inferred from chimeric reads; UD, undetermined

LENG9 was significantly up-regulated upon HPV integration in tumor ZLR-08, and its expression level was 16-fold higher than its adjacent normal tissue. To evaluate the potential functional role of LENG9 in cervical carcinoma, we overexpressed LENG9 in cervical cancer cell lines CaSki and SiHa (new **Fig. 6C**). For the overexpression assay, a full-length human LENG9 sequence was obtained by PCR and subcloned into the PLVX-puro vector by Lipofectamine 3000 to establish cells that stably overexpressed LENG9. Consistently, upregulation of LENG9 in CaSki and SiHa cells significantly increased proliferation, migration and invasion of cervical cancer cells (new **Fig. 6D-F**). In addition, we performed knockdown assays by treating CaSki cells with siRNAs targeting LENG9. As a result, depletion of LENG9 significantly inhibited cell proliferation, migration and invasion of cervical cancer cells (new **Fig. S6**). Taken together, these results suggest that LENG9 may play an oncogenic role in tumor pathogenesis through promoting tumor cell proliferation, migration, and invasion.

In summary, HPV-mediated amplification of host genome fragments with cis-activator of oncogene or oncogene *per se* could be a potential mechanism for cervical carcinogenesis and progression. Thank you again for pointing this out and we have corrected the relevant text, tables and figures in the revised manuscript (pages 9-10).

4. The authors mentioned "substantial intra-tumor heterogeneity" in the abstract section. However, I could not find sufficient explanations of clone patterns and impact of heterogeneity to oncogenic progression and patient prognosis in cervical cancer. I think that, for example, variant allele frequency of integration breakpoints and surrounding mutations in the host genome would be informative for understanding clone architecture.

Response:

Thank you for pointing out this important issue! The intratumor heterogeneity generally refers to the existence of distinct cellular populations within the same tumor. In short-read bulk sequencing studies, the intratumor heterogeneity can be indirectly inferred from somatic allelic mutation data and copy-number aberration data. These previous studies sought to demonstrate whether different cell populations with different gene mutation patterns and copy number variation patterns exist within the same tumor. Instead, the present study aimed to explore the potential intratumor

heterogeneity in the context of the distribution of integration events. In other words, we sought to demonstrate whether different cell populations carry different integration events within the same cervical tumor.

In our study, the potential existence of intratumor heterogeneity of HPV integration in cervical cancer was inferred based on the following observations: First, 3 of 16 samples (i.e., ZLR-08, ZLR-11 and ZLR-12) detected three or more clonal integration events that were defined as a collection of chimeric reads (> 5 kb) containing at least one fragment of HPV DNA wrapped on both sides by the human genome. These distinct integration events in the same sample tended to be clustered together and partially overlapped with different breakpoints. Nanopore technology is an amplification-free sequencing method, allowing direct, long-read sequencing of native DNA. Each chimeric read likely represents a single cell within the sequenced bulk cells. It is less likely that these integration events simultaneously occurred in a diploid cell since there are generally only two copies of a specific chromosomal site for viral integration. Second, we observed that shared breakpoints are being used in different integration events within the same tumor. It is also less likely that these distinct integration events occurred in different cells independently yet producing the same breakpoint (new **Fig. 4**). A reasonable inference is that these integration events are evolutionarily related, and some integration events may derive from a primary integration event in a subset of its clonal cells.

To further demonstrate evidence of intratumor heterogeneity of HPV integration in cervical tumors, below we presented a schematic representation of three captured authentic chimeric reads that represent three different integration events in ZLR-12 (**Fig. R1**). The junction site ① is the shared site of three distinct integration events and its left stretched sequence in these integration events were captured by long-read sequencing. Since both the normal flanking human chr17 sequence (in e1 and e2) and the truncated and stitched hybrid sequence (in e3) are present, these three distinct integration events are unlikely occurred in the same cell due to the diploid nature of normal human cells. Furthermore, the sequence depth reported by sniffles for junction sites ①, ②, and ④ are 58, 44, and 18, respectively. This suggested that the junction site ④ occurred less frequently among these junction sites, even though it was tandemly duplicated in ZLR-12_e3. A plausible explanation is that (i) the shared breakpoint ① may originate from the ancestral clonal integration event, in which the first viral integration occurred in cervical epithelial cells; (ii) these integration events sharing the common breakpoint ① are likely evolutionarily related; and (iii) the integration event ZLR-12_e3 may be a later event during cervical cancer progression. These observations provide evidence that these integration events may be involved in clonal evolution, thereby contributing to another layer of intratumor heterogeneity in cervical cancer in addition to somatic mutations and copy number alterations.

ZLR-12

Figure R1. A schematic illustration of three captured authentic chimeric reads that represent three different integration events in ZLR-12.

As for the impact of intratumor heterogeneity on patient prognosis, we collected the follow-up information of 16 patients (revised **Table 1**) as you previously suggested. Surprisingly, two patients (ZLR-11 and ZLR-12) with multiple integration events displayed some intratumor heterogeneity and had a poor prognosis (decease). In revised dataset, the ZLR-08 could also be inferred with intratumor heterogeneity based on the commonly used breakpoints among her five integration events. However, we lost any contact of ZLR-08 patient, who is neither on the list of compulsory vaccination against COVID-19 in my country nor on the list of free gynecological tumor screening program, presumably suggesting a poor prognosis.

In summary, we inferred the potential existence of intratumor heterogeneity in three samples (ZLR-08, ZLR-11 and ZLR-12) based on the observation of commonly used breakpoints among multiple integration events (>2) within the same tumors. Furthermore, adverse prognosis was observed only for these patients, strongly suggesting the existence of intratumor heterogeneity and its association with worse prognosis. According to your suggestion, the number of high-quality reads supporting the reported breakpoints was extracted and added to the revised **Table S3**. We also added more explanations on intratumor heterogeneity and its impact on patient prognosis and further discussed it in the revised manuscript (pages 7 and 13). Thank you very much again to point this out.

Minor points;

1. In Fig. 2A, arrows are out of position.

Response:

Thank you for your careful check. We have adjusted the arrows to the correct position in the revised Figure 2A.

Reviewer #2 (Remarks to the Author): Expert in HPV and cervical cancer genomics

General Comments:

The Nanopore long-read sequencing technology has opened new avenues in research related to HPV-DNA integration. Zhou and colleagues present a well-written paper highlighting novel findings which are clearly of interest for experts in this field. Largely the conclusions drawn are supported by the data provided but several aspects require clarification:

Response:

Thank you very much for your comments! We have addressed your comments one by one below.

1. Page 2, lines 54 to 56 state: “Our study depicts large-range virus-human integration events at high resolution with several novel characteristics and reveals substantial intra-tumor heterogeneity of HPV integration in cervical carcinogenesis.” In this sentence, the term intratumor heterogeneity is misleading. The authors need to define in their manuscript what is actually meant by intratumor heterogeneity. As reported in several previous papers most tumors harbor more than one HPV integration event. If this is meant by intratumor heterogeneity their findings would be of confirmative nature. It would, however, be of interest to know whether the integration events have occurred within the same tumor cell or are scattered throughout the tumor giving rise to different subclonal tumor cell populations. The authors provide no evidence for this type of intratumor heterogeneity. Strong evidence for the lack of intratumor heterogeneity in terms of distribution of integration events throughout the tumor tissue is provided by Carow et al. 2017 (doi:10.3390/ijms18102032). These findings need to be discussed.

Response:

Thank you for the important reminder and we apologize for the inaccurate wording of “substantial intratumor heterogeneity” in the manuscript. We agree with the latter definition that intratumor heterogeneity generally refers to the existence of distinct cellular populations within the same tumor. We thus explored the potential intratumor heterogeneity in the context of the distribution of integration events. For 12 samples with detection of HPV integration events, 8 tumors harbored 1 integration event and 4 tumors harbored multiple integration events (revised **Table 1**). For tumors harboring single integration event, it is tempting to speculate their monoclonal origin and intratumor homogeneity, which is also supported by micro-dissection analyses of 4 tumors with single HPV integration by Carow and colleagues (*Int J Mol Sci.* 2017, 18(10): 2032).

In our study, taking advantage of long-read sequencing, integration events were defined as a collection of chimeric reads (> 5 kb) containing at least one fragment of

HPV DNA wrapped on both sides by the human genome. The potential existence of intratumor heterogeneity in cervical cancer was inferred in 3 of 16 samples (i.e., ZLR-08, ZLR-11 and ZLR-12) that harbored at least three integration events. First, these distinct integration events in the same tumor tended to be clustered together and partially overlapped with different breakpoints. Nanopore technology is an amplification-free sequencing method, allowing direct, long-read sequencing of native DNA. Each chimeric read likely represents a single cell within the sequenced bulk cells. It is less likely that these integration events simultaneously occurred in a diploid cell since there are generally only two copies of a specific chromosomal site for viral integration. Second, we observed that shared breakpoints are being used in different integration events within the same tumor. It is also less likely that these distinct integration events occurred in different cells independently yet producing the same breakpoint (new **Fig. 4**). A reasonable inference is that these integration events are evolutionarily related, and some integration events may derive from a primary integration event in a subset of its clonal cells.

To further demonstrate evidence of intratumor heterogeneity of HPV integration in cervical tumors, below we presented a schematic representation of three captured authentic chimeric reads that represent three different integration events in ZLR-12 (**Fig. R1**). The junction site ① is the shared site of three distinct integration events and its left stretched sequence in these integration events were captured by long-read sequencing. Since both the normal flanking human chr17 sequence (in e1 and e2) and the truncated stitched hybrid sequence (in e3) are present, these three distinct integration events are unlikely occurred in the same cell due to the diploid nature of normal human cells. Furthermore, the sequence depth reported by sniffles for junction sites ①, ②, and ④ are 58, 44, and 18, respectively. This suggested that the junction site ④ occurred less frequently among these junction sites, even though it was tandemly duplicated in ZLR-12_e3. A plausible explanation is that (i) the shared breakpoint ① may originate from the ancestral clonal integration event, in which the first viral integration occurred in cervical epithelial cells; (ii) these integration events sharing the common breakpoint ① are likely evolutionarily related; and (iii) the integration event ZLR-12_e3 may be a later event during cervical cancer progression. These observations provide evidence that these integration events may be involved in clonal evolution, thereby contributing to another layer of intratumor heterogeneity in cervical cancer in addition to somatic mutations and copy number alterations.

ZLR-12

Figure R1. A schematic illustration of three captured authentic chimeric reads that represent three different integration events in ZLR-12.

In summary, we found that 3 out of 16 tumors displayed potential intratumor heterogeneity, which is comparable to the work of Carow and colleagues in which micro-dissection analysis revealed 1 out of 8 analyzed tumors exhibiting intratumor heterogeneity (ref 50). We restated the exaggeration of “substantial intratumor heterogeneity” in the related text, added detailed explanation on intratumor heterogeneity in the discussion in the revised manuscript (pages 7 and 13).

2. Page 3, lines 77 to 79 state: “Upon integration, overexpression of viral oncogene E6 and/or E7 is frequently observed, which may result from the disruption of their negative regulators E1/E2, and the increased stability of E6 and/or E7 mRNAs expressed from the viral-host fusion transcripts.” Häfner et al. 2008 (doi: 10.1038/sj.onc.1210791) provide evidence that HPV16 DNA integration does not invariably result in high levels of oncogene transcripts. Moreover, the transcript levels between CIN (mainly high-grade lesions) and cervical carcinomas were shown to be similar. Concerning transcript stability the work by Ehrig et al. 2020 (doi:10.3390/ijms21010112) provides further insights and should be cited.

Response:

Thank you for pointing out this potentially controversial issue. In the study of Häfner et al., APOT-assay was employed to determine the physical state of HPV DNA in 28 CIN and 55 cervical carcinomas, and to further evaluate if the expression levels of viral oncogene transcripts (measured by qRT-PCR) were correlated with the viral physical state in these samples. It turned out that there was no significant difference in the E6/E7 transcript levels among three different viral physical state groups (i.e., episomal, integrated, and a mix of the two). This result appeared to be contrary to a previous *in vitro* study using W12 cell populations (JEON et al. 1995, DOI: 10.1158/1055-9965.EPI-04-0410). In that study, clonal populations with integrated viral DNA had increased levels of viral E7 protein and cell outgrowth even with much lower HPV copy number compared with clonal populations harboring extrachromosomal viral DNA, and they concluded that HPV integration correlates

with increased viral gene expression and cellular growth advantage. A possible explanation for this discrepancy, as the authors mentioned, is sample source (biopsies *versus* cell lines). Another possible explanation is that in addition to HPV integration, deregulated expression of E6/E7 oncogene can also be achieved by other ways such as genetic or epigenetic changes where HPV can still remain as episomal forms (*Dong et al. 1994, DOI:10.1002/ijc.2910580609; Bhattacharjee et al. 2006, DOI:10.1016/j.virol.2006.06.018*). Lastly, a technical limitation of APOT-assays to classify HPV physical states is that it may oversight integration events with disruption in L1/L2 genes or with the form of concatemers, although not very common, and thus may influence the outcomes. For our dataset, all biopsy samples except one sample (ZLR-05) contain integrated HPV DNA and fusion transcripts, thus these samples are not informative for similar analysis.

We also referred to more other literatures on this topic and realized “overexpression” was improperly phrased, and “deregulated/continuous/constitutive/altered expression” would be more appropriate. We have rephrased the relevant text and discussed the work by Häfner et al. 2008 (doi: 10.1038/sj.onc.1210791) in the revised manuscript (page 3 and ref 18). In addition, the insightful work concerning transcript stability by Ehrig et al. 2020 (doi:10.3390/ijms21010112) was properly cited in the revised manuscript (page 3 and ref 17).

3. Page 3, lines 90 to 91 state: “Using NGS technologies, several studies have been carried out to characterize genome-wide alterations in both viral and host genomes upon HPV integration in cervical cancer.” A seminal contribution for the detection of HPV integration sites using NGS technology was also made by Xu et al. 2013 (doi: 10.1371/journal.pone.0066693) and should be cited.

Response:

We gratefully appreciate for your suggestion. We have properly cited this article in the revised manuscript (ref 31).

4. Page 8, lines 230 to 232 state: “Genes at HPV hot spot integration sites included KLF5, LINC00392, CASC8, CASC21, LINC00290 and MACROD2. In particular, the KLF5-LINC00392 on chromosome 13 was identified at a recurrent integrated site with 5 breakpoints in two tumors.” The term hot spot is misleading. In previous publications the term “hot spot” refers to common chromosomal regions which are affected by HPV integration in several independent tumors (see Schmitz et al 2012, doi: 10.1371/journal.pone.0039632 and Hu et al 2015, doi:10.1038/ng.3178). This then poses the question as to which integration events of the current study map to previously reported hot spots?

Response:

We are sorry for the confusion. Due to the lack of full understanding on the term “hot spot”, in our original version, we not only presented KLF5, LINC00392, CASC8, CASC21 and MACROD2 genes which meet the reported hot spots to our knowledge but also included LINC00290 that we consider important. Thanks to your comments, we now have a better understanding on this term. Based on our own data set, except for the region between KLF5 and LINC00392 found in two tumors, we could not find other HPV integration hot spots largely due to the limited sample size. Following your suggestion, we collected several publications containing information on HPV integration hot spots and mapped our results to these previously reported hot spots. As a result, KLF5, LINC00392, CASC8, CASC21, MACROD2, TEX41 and VMP1 (alias TMEM49) were mapped to previously reported hot spots (Schmitz et al 2012, doi: 10.1371/journal.pone.0039632; Li et al 2018, doi: 10.1155/2018/6242173; Kamal et al 2020, doi: 10.1038/s41416-020-01153-4). We have corrected the relevant text in the revised manuscript (page 9 and ref 38-40).

5. Page 9, lines 253 to 254 state: “We particularly focused upon two HPV-integrated tumors (ZLR-01 and ZLR-08) with low expression of E6 and E7 whose upregulation had been considered to be primarily responsible for cervical carcinogenesis.” Figure 5A and 5B show the expression level of E6 and E7 of 103 cervical carcinomas, respectively. Expression in sample ZLR-08 is highlighted by a red dot. It is rather surprising that a large proportion of cervical cancers examined have such low levels of E6/E7 RNA. There seems to be a 6 fold log₂ difference in transcription levels. This is not in line with the general understanding that cervical cancers have high E6 and E7 transcript levels. This should be explained.

Response:

Thank you for your comments. Many patient samples were affected with multiple HPV types. In the original figure 5A and 5B, we only focused on the expression of HPV16 E6/E7 and did not consider the expression of E6 and E7 in other HPV types. To avoid the confusion, we included different HPV types, and evaluated the total E6/E7 expression of corresponding infected HPV types in each sample. The new analysis showed that only 8 of 103 cervical tumor samples had low levels of E6/E7 expression (TPM of E6 and E7 less than 3), which is more in line with the general findings of previous cervical cancer studies (**Table 1** and **Fig. S7**).

6. Page 12, lines 379 to 382: “Of note, Type B is a novel type of integrated HPV DNA segment in the host genome, which lacks E6/E7 genes that have been previously considered to be the main causes for cervical cancer. Despite this, Type B was common, observed in 2 of the 12 tumors detected with clonal integration events.”

Can “2 of 12 tumors” be considered to be a common event? Moreover, Type B integrated fragments do not seem to be the only integration event in the respective tumors. These tumors seem to also harbor E6/E7 containing integration events (see

Figure 3).

Response:

Thank you for pointing out this improper expression. To our knowledge, integration Type B was rarely reported in the literature while in our limited sample size it occurred three times across two tumors (there were five Type B integration events across three tumors in our revised dataset), which was beyond our expectation. Furthermore, the DNA depth/concentration of Type B associated integration event (ZLR-11_e1) was highest in ZLR-11 (**Fig R2**). On the other hand, as you remarked, the Type B integration event was not exclusively presented in a tumor based on our dataset. This indicates that the Type B integration event may be a “passenger event” during cervical disease progression. However, it is also possible that the Type B integration event may co-amplify with other oncogenes of host genome (such as CCAT2 in ZLR-11), thereby promoting cervical carcinogenesis. These remain to be further studied. We have rephrased our improper expression and discussed this new type of event in the revised manuscript (page 14).

Fig. R2 DNA sequence base coverage of Type B associated integration event (ZLR-11_e1).

7. Page 13; lines 387 to 389 state: “Overexpression of E6/E7 genes upon HPV integration is frequently observed in HPV-associated cancers. It is this that has been thought to be primary trigger of the oncogenic progression by disturbing the cell cycle and inducing genomic instability in cervical cancer.”

As already mentioned above, DNA integration does not invariably result in high levels of oncogene transcripts. Moreover, the transcript levels between CIN (mainly high-grade lesions) and cervical carcinomas were shown to be similar. Concerning

transcript stability the work by Ehrig et al. 2020 (doi:10.3390/ijms21010112) provides further insights and should cited.

Response:

We agree with you that HPV integration does not invariably result in high levels of oncogene transcripts. As you suggested, we rephrased the relevant text and cited the relevant work in the revised manuscript (page 14 and please also see our response to the comment #2 above).

REVIEWERS' COMMENTS

Reviewer #1 (Remarks to the Author):

In this revised manuscript, the authors showed the follow-up information of the patients and described clear association between each feature of integration events and clinical consequences of cancers. This information would support to interpret the results.

They newly focused on a gene LENG9 and added experiments for functional relevance of this gene to cervical cancer pathogenesis. I have some additional comments about this gene as below;

Points;

1. Please investigate whether LENG9 expression is associated with patient prognosis by using TCGA or other large cohort data in a similar manner to the previous analysis of CDC42EP5.
2. The authors should more specifically discuss the potential function and the related pathways of LENG9 in cervical cancer even though this gene is rarely reported in the literature.

Miscellaneous;

1. The authors newly advocated that LENG9 gene was a potential driver gene in cervical cancer instead of CDC42EP5. According to the original and revised version of the manuscript, I think that CDC42EP5 gene is still one of the candidates for cause of oncogenic events in cervical cancer because aberrant genomic events also affected CDC42EP5 locus. Please clearly explain the authors' opinion whether CDC42EP5 could be associated with cervical cancer pathogenesis or not.
2. Line 616: "Primers of CDC42EP5 for quantitative RT-PCR" was shown. Please provide the information for LENG9 gene if the authors did the experiment for this gene.

Reviewer #2 (Remarks to the Author):

In their revised version of the manuscript the authors have fully addressed all criticisms raised by the reviewer. The authors provide comprehensible novel insights pertinent to the field of HPV-DNA integration and cervical carcinogenesis. Congratulations!

Below are responses to comments from Reviewer #1. We greatly appreciate his/her additional thoughtful comments and suggestions, which further improved and clarified the presentation of our manuscript.

Please note that *the comments are shown in italics and bold font*. Figures R1-R4 are new displayed items for reviewers only that were generated from our recent data analyses in response to reviewers' comments. The other figures referred to in the responses can be found in either the main text or supplementary data of our manuscript. Changes made in the revised manuscript were highlighted in red font.

Reviewer #1 (Remarks to the Author):

In this revised manuscript, the authors showed the follow-up information of the patients and described clear association between each feature of integration events and clinical consequences of cancers. This information would support to interpret the results. They newly focused on a gene LENG9 and added experiments for functional relevance of this gene to cervical cancer pathogenesis. I have some additional comments about this gene as below;

Response:

We highly appreciate the time and effort that you have dedicated to providing your valuable comments on our manuscript. We have addressed your additional comments one by one below.

Points;

1. Please investigate whether LENG9 expression is associated with patient prognosis by using TCGA or other large cohort data in a similar manner to the previous analysis of CDC42EP5.

Figure R1 Kaplan-Meier curves of survival of cervical cancer patients with high versus low expressions of LENG9 in TCGA. (A) Overall survival (OS). (B) Disease-specific survival (DSS). Patients with cervical cancer were divided into two groups according to the median expression of LENG9. Differences in survival between two groups were evaluated using the log-rank test.

Response:

Thank you for your valuable suggestion! Our investigation did not find a significant association of LENG9 expression with prognosis of cervical cancer patients in TCGA (**Figure R1**). We also searched other GEO datasets of cervical cancer, but these datasets lacked survival information and could not be used for our investigation. It is worth noting that one of the two studies found in PubMed reported that a six-RBP gene signature (including LENG9) is predictive of survival of prostate cancer patients (*Genomics* 2020, 112:4980-4992). The prognostic value of LENG9 expression in cervical cancer remains to be further investigated in other large datasets to draw firm conclusions.

2. The authors should more specifically discuss the potential function and the related pathways of LENG9 in cervical cancer even though this gene is rarely reported in the literature.

Response:

Thank you for your valuable comments. Following your suggestion, we further explored the potential function and related pathways of LENG9. GO function analysis revealed that its molecular function is metal ion binding (GO:0046872) with zinc finger domain ZnF_C3H1. The BioGRID (*Nucleic Acids Res.* 2006, 34(suppl_1): D535-D539), a curated database for protein, genetic and chemical interactions, showed that LENG9 has 6 interactors, including C9ORF41, B4GALT2, CDC5L, D2HGDH, FOXS1 and OGT (**Figure R2**; <https://thebiogrid.org/125106>). Among them, B4GALT2, FOXS1 and OGT were found to be related with cancer. In particular, B4GALT2 is also annotated to GO term of metal ion binding (GO:0046872) and its expression is dramatically reduced in tumor ZLR-08 compared to adjacent normal tissue. It has been reported that depletion of B4GALT2 inhibits p53-mediated cell apoptosis in Hela cells (*J Biochem.* 2008, 143(4):547-54). Furthermore, CDC5L (alias HCDC5) has been reported as a positive regulator of cell cycle G2/M progression (*J Biol Chem.* 1998, 273(8): 4666-4671). Their interactions with LENG9 may be a potential mechanism underlying the promotion of cell proliferation and migration when LENG9 is upregulated in cervical cancer cells while these warrant further functional investigation in the future (pages 14-15).

Figure R2 Six genes/gene products potentially interacting with LENG9. This image was captured from the network overview of LENG9 interactions in BioGRID v4.4.

Miscellaneous;

1. The authors newly advocated that LENG9 gene was a potential driver gene in cervical cancer instead of CDC42EP5. According to the original and revised version of the manuscript, I think that

CDC42EP5 gene is still one of the candidates for cause of oncogenic events in cervical cancer because aberrant genomic events also affected CDC42EP5 locus. Please clearly explain the authors' opinion whether CDC42EP5 could be associated with cervical cancer pathogenesis or not.

Response:

Thank you for your rigorous consideration. We claimed LENG9 as a primary candidate in the HPV integration region of ZLR-08, in part because the expression of CDC42EP5 was not significantly different between ZLR-08 and other tumors evaluated in our study (**Fig. 6B**). However, we agree with you that CDC42EP5 may be secondary candidate for the oncogenic event in ZLR-08 because the aberrant genomic event also affected CDC42EP5 locus.

Figure R3 Transcript read coverage plot for CDC42EP5 in ZLR-08. Based on RNA-seq data, the read coverage curves spanning exons of CDC42EP5 transcript are displayed in red and blue for tumor and normal of ZLR-08, respectively. The structures of CDC42EP5 transcripts with scaled introns are shown below the curves. RPB: the number of reads per bin (50 bp).

As shown in Figure 6A, CDC42EP5 was partially (two out of three exons) amplified upon HPV integration. This effect was also projected onto the corresponding transcripts, in which only the last two exons of CDC42EP5 were extensively transcribed (**Figure R3**). However, only the last exon was shown to encode a functional protein (**Figure R4**), suggesting that the function of CDC42EP5 may be partially preserved although it was partially amplified.

CDC42EP5, also named as Borg3, is a protein coding gene that encodes a member of the Borg (binder of Rho GTPases) family of CDC42 (cell division control protein 42) effector proteins. The revealed biological functions of this protein include cell shape regulation, inducing pseudopodia formation in fibroblasts, increasing thin stress fibers and reducing E-cadherin expression at adherens junctions in

keratinocytes; this suggests that CDC42EP5 may play a role in epithelial-to-mesenchymal transition (EMT) and cancer metastasis (*J Biol Chem.* 2001, 276(2):875-883; *Biochem Soc Trans.* 2016, 44(6):1709-1716). A recent study also reported that this gene plays a positive regulatory role in melanoma migration, invasion and metastasis by coordinating actin and septin networks (*J Cell Biol.* 2020, 219(9): e201912159.). Our previous functional assays in cervical cancer cell lines CasKi and C-4I also verified the function of CDC42EP5 in promoting cell migration and invasion.

In our case, the protein-coding exon region (i.e., exon 3) of CDC42EP5 is completely amplified and extensively transcribed. Given the fact that the part of a gene containing core domain can still exert its function in some cases (e.g., the tyrosine kinase domain of ALK in EML4-ALK fusion gene; *Nature.* 2007, 448:561-566), it is likely that in our case, CDC42EP5 may be involved in the pathogenesis of cervical cancer, but it remains to be further studied. **Therefore, we added a brief discussion on the candidacy of CDC42EP5 in the revised manuscript (page 15).**

Figure R4 Gene annotation of CDC42EP5 (highlighted with light green background) in the Ensemble Genome Browser (hg19). The canonical transcript of CDC42EP5 has three exons, and only the last exon encodes the protein. This image was captured from Ensembl GRCh37 release 105 (*Nucleic Acids Res.* 2021, 49(1):884-891).

2. Line 616: "Primers of CDC42EP5 for quantitative RT-PCR" was shown. Please provide the information for LENG9 gene if the authors did the experiment for this gene.

Response:

Thank you for your reminder. In the revised manuscript, we provided the primer information for quantitative RT-PCR of LENG9 in Table S7. To avoid confusion, we deleted the related text on primer information of CDC42EP5 in the Method section.